# Beyond Pharmacology: A Narrative Review of Alternative Therapies for Anxiety Disorders

**DOI:** 10.3390/diseases12090216

**Published:** 2024-09-16

**Authors:** Zuzanna Antos, Klaudia Zackiewicz, Natalia Tomaszek, Stefan Modzelewski, Napoleon Waszkiewicz

**Affiliations:** Department of Psychiatry, Medical University of Bialystok, pl. Wołodyjowskiego 2, 15-272 Białystok, Poland; 39615@student.umb.edu.pl (Z.A.); 38791@student.umb.edu.pl (K.Z.); 38778@student.umb.edu.pl (N.T.); napoleon.waszkiewicz@umb.edu.pl (N.W.)

**Keywords:** alternative treatment, anxiety, phobia, anxiety disorders, PTSD, physical activity, VR, herbs, yoga, mindfulness

## Abstract

Background: Anxiety disorders significantly reduce patients’ quality of life. Current pharmacological treatments, primarily benzodiazepines and antidepressants, are associated with numerous side effects. Consequently, there is a continual search for alternative methods to traditional therapies that are less burdensome for patients and broaden their therapeutic options. Our objective was to determine the role of selected alternative methods in the treatment of anxiety disorders. Methods: In this review, we examined recent evidence on alternative treatments for anxiety disorders, including physical activity, mindfulness, virtual reality (VR) technology, biofeedback, herbal remedies, transcranial magnetic stimulation (TMS), cryotherapy, hyperbaric therapy, vagus nerve stimulation (VNS), 3,4-methylenedioxymethamphetamine (MDMA), electroconvulsive therapy (ECT), and eye movement desensitization and reprocessing (EMDR) therapy. For this purpose we reviewed PubMed and after initial search, we excluded works unrelated to our aim, non-orginal data and animal studies. We conducted second search to cover all minor methods. Results: We included 116 studies, which data is presented in Tables. We have investigated which methods can support treatment and which can be used as a stand-alone treatment. We assessed the risks to benefits of using alternative treatments. Conclusion: Alternative treatments significantly expand the options available to patients and clinicians, with many serving as adjuncts to traditional therapies. Among the methods presented, mindfulness has the most significant therapeutic potential.

## 1. Introduction

Anxiety disorders (ADs) present a significant global public health challenge, impacting an estimated 275 million individuals annually and giving rise to approximately 42 million new cases each year [1]. In Europe, ADs exhibit a prevalence rate of 14.0%, the highest among all mental disorders [2]. ADs profoundly affect individual functioning, impeding social interactions, causing sleep disturbances, and elevating the likelihood of somatic symptoms such as pain or fatigue. Moreover, they predispose individuals to other psychiatric disorders, substance use disorders, and physical ailments, including cardiovascular disease [3].

Individuals diagnosed with ADs frequently seek medical services, leading to increased healthcare costs and economic implications due to reduced workplace productivity [4]. A range of treatment options is available for managing ADs, such as cognitive-behavioral therapy and pharmacotherapy, primarily involving antidepressants and benzodiazepines. However, despite their proven efficacy, these medications carry significant risks of adverse effects. Prolonged benzodiazepine use can cause addiction, while abrupt discontinuation may trigger withdrawal syndrome. Similarly, prolonged use of selective serotonin reuptake inhibitors (SSRIs) can lead to various adverse effects [5,6]. It is important to note that the side effects of anxiolytic medications can outweigh their potential benefits, especially in individuals with mild clinical presentations. For example, patients with anxiety disorders may encounter medication-induced side effects that exacerbate their anxiety, decrease adherence to the prescribed treatment, and ultimately lead to poorer therapeutic outcomes [7,8]. The overuse of healthcare services, including unnecessary diagnostic testing, can also lead to heightened anxiety and additional side effects, underscoring the importance of considering alternative treatment approaches. 

Moreover, the use of antianxiety medications in pregnant women with anxiety disorders poses significant risks. Concerns about medication safety during pregnancy can lead to self-discontinuation of drugs, potentially resulting in adverse pregnancy outcomes [9]. Thus, it is crucial to carefully weigh the benefits and risks of antianxiety medications in this population and consider safer alternative treatments.

In addition to pregnant women, other vulnerable populations, such as adolescents and the elderly, may benefit from alternative anti-anxiety treatment strategies. Adolescents often experience stressors related to the school environment and the challenges of youth. Managing these stressors does not always require pharmacological methods, which could interfere with daily functioning. Therefore, exploring non-pharmacological options such as cognitive-behavioral therapy (CBT) and mindfulness practices seems to have great potential in this patient group. Similarly, older adults often face comorbidities that complicate anxiety treatment, making it essential to evaluate the effectiveness of alternative therapies, including psychotherapy and lifestyle interventions, to improve their overall mental health [10,11]. By broadening the focus to these populations, we can better address the complexities of treating anxiety and improve outcomes for different patient groups. 

Despite the presence of numerous treatment options, significant barriers hinder the widespread implementation of cognitive-behavioral therapy (CBT). These obstacles include a shortage of therapists, social stigma, prolonged waiting periods for treatment, and the financial burden associated with therapy [12]. 

What is worth mentioning is that a considerable portion of individuals with mood disorders either do not receive treatment or receive ineffective assistance [13]. Hence, alternative therapeutic approaches are under exploration, aiming to better align with individual patient preferences. These methods are not only more cost-effective but also less prone to causing side effects, rendering them appealing treatment options.

## 2. Materials and Methods

The aim of our study was to provide a narrative review of research on the use of alternative methods in anxiety disorders, with a particular focus on generalized anxiety disorder (GAD) and social anxiety disorder (SAD). Additionally, our review was expanded to include selected methods related to post-traumatic stress disorder (PTSD), which in DSM-V is classified separately, due to the dynamic development of PTSD treatment methods, which are subsequently applied to anxiety disorders. It is important to note that our work is not systematic and represents a narrative review to assess the topic broadly. By reviewing the selected articles, we aimed to broadly describe alternative treatments and answer the following questions:What alternative treatment methods can serve as alternatives to conventional treatments?What methods can serve only as supplements to traditional treatments?What are the typical difficulties associated with conducting these types of therapies?

The selection of articles and basic data concerning them are presented in Appendix A for physical activity, Appendix A for the use of VR, Appendix A for Mindfulness, Appendix A for neurostimulation, Appendix A for biofeedback, and Appendix A for herbal treatments. Additionally, after expert consultation, we conducted a subsequent database search, expanding the scope to include methods such as barotherapy, cryotherapy, vagus nerve stimulation (VNS), eye movement desensitization and reprocessing (EMDR) in PTSD, and 3,4-methylenedioxymethamphetamine (MDMA) in the context of the studied disorders, which are presented in Appendix A.

For the initial search, three authors independently searched the PubMed database, which included works exclusively in English, only original data, from the years 2002 to 2024*. During the search, we used the keywords combined from “GAD”, “phobia”, “SAD”, and “PTSD” with “physical activity”; “yoga”, “resistance training”, “VR”, “mindfulness”, “TMS”, “herbs”, and “treatment”. The search process is schematically illustrated in a flowchart (Figure 1). The database review was conducted in April 2024, and we collected 439 records. We removed duplicates using EndNote version 21 software. The second search was conducted in the same way in May 2024.

Next, the prepared database was reviewed by examining titles and abstracts. The review was conducted independently by three authors. We excluded works unrelated to our research questions, and animal studies. Finally, our narrative review included 116 studies. From each study, we extracted data regarding study characteristics (authors, year, additional details), limitations, patient count, and key findings. Data are presented in Appendix A for the first search and in Appendix A for the second search. Tables are available in Appendix A. We presented the results of our narrative review as a set of chapters describing the following identified alternative treatments in sequence: physical activity, VR technology, mindfulness, neurostimulation (we mainly focused on TMS), biofeedback, and other methods, such as EMDR therapy, vagus nerve stimulation, adjuvant MDMA treatment, cryotherapy, barotherapy, and herbal treatment.

## 3. Physical Exercise 

Regular physical exercise offers numerous positive health benefits, including reduced overall mortality, improved musculoskeletal health, and enhanced stress regulation, as well as shows potential to regulate mental health [14]. Research suggests that physical activity modulates 5HT 1a and 5HT 2c receptors in limbic structures of the central nervous system. These neurobiological changes may contribute to a reduction in symptoms of panic disorder and an enhancement of overall mental well-being among patients [15]. Additionally, individuals with high baseline anxiety levels in response to moderate exercise experienced a heightened reduction in perceived anxiety with each successive week [16] of exercise. The primary categories of physical activity that have shown significant research potential are yoga, resistance, and aerobic exercises. 

Practicing yoga can reduce symptoms of anxiety disorders by regulating the HPA (hypothalamic–pituitary–adrenal) axis and increasing the level of brain-derived neurotrophic factor (BDNF), which supports neuroplasticity. Moreover, an increasing number of studies indicate that practicing yoga influences the modulation of inflammatory markers, which seems to be related to the pathogenesis of anxiety disorders [17]. A study involving healthy individuals revealed that a 3-month yoga therapy regimen increased thalamic gamma-aminobutyric acid (GABA) levels and decreased anxiety [18]. Furthermore, a 2-month yoga therapy intervention in women with anxiety disorders resulted in a significant reduction in feelings of anxiety, suggesting the potential use of yoga as an adjunct to traditional drug therapy [19]. Integrating yoga with CBT accelerated the reduction of anxiety and depressive symptoms compared to using CBT alone. The therapeutic effect persisted for up to 3 months at follow-up [20]. Lundt et al. observed that yoga appeared to be a beneficial adjunct to cancer treatment, significantly reducing symptoms of anxiety, depression, and fatigue six months post-treatment. This shows that yoga can be effective in treating anxiety associated with anxiety disorders as well as in treating anxiety associated with all kinds of difficult life situations [21]. 

Broman-Fulks et al. showed that individuals with heightened anxiety levels experienced a significant reduction in anxiety and sensitivity when engaging in high-intensity exercise compared to low-intensity exercise. Additionally, the anxiolytic effects were noticeable as early as the outset of the second workout, with an average interval of 2 days between sessions. The reduction in anxiety achieved through high- and low-intensity exercises persisted for up to a week post-intervention [22]. 

In another study, Broman-Fulks et al. [23] reported no clinically significant reduction in anxiety levels measured 5 min after physical activity, consistent with findings from the Cox et al. study [24]. The latter suggests a delayed anxiolytic effect of aerobic exercise, becoming noticeable only after 30 min. Additionally, they observed greater tolerance of the CO_2_ test among participants engaged in aerobic exercise compared to those in resistance exercise groups. Nevertheless, both exercise groups exhibited lower anxiety levels than the non-exercise group. 

In a distinct study, which included participants with various anxiety disorders, such as panic disorder (PD), specific phobias, PTSD, GAD, and specific phobias, notable findings emerged. Following a three-month follow-up, during which individuals engaged in regular exercise sessions five times a week, each lasting 30 min, researchers noted a significant reduction in perceived anxiety. These findings suggest that the heightened self-awareness accompanying physical conditioning may empower individuals to perceive enhanced control over their health, thereby potentially alleviating anxiety [25]. As mentioned above, there is a significant amount of research and evidence pointing to the validity of evaluating the impact of different types of physical activity on the treatment of anxiety disorders.

### 3.1. GAD

#### 3.1.1. Physical Exercise as an Adjuvant to Pharmacotherapy

Gordon et al. explored whether physical activity, particularly resistance exercise training (RET), can delay or prevent the onset and progression of an analog generalized anxiety disorder (AGAD). Early intervention is crucial as it can significantly reduce the likelihood of future clinical psychopathology. The findings support the recommendation of guideline-based aerobic and RET as effective therapies for anxiety and worry symptoms. RET, designed according to WHO and ACSM (American College of Sports Medicine) guidelines, significantly improved AGAD status among young adults. The number needed to treat (NNT) was three, indicating that one in three participants would achieve remission through this intervention. This NNT is superior compared to antidepressant treatment and similar to cognitive behavioral therapy for GAD [26]. Higher-intensity interval training (HIIT) in patients with GAD resulted in twice the anxiety reduction compared to lower-intensity exercise (LIT) [27]. Another randomized study indicated a 60% remission of symptoms in GAD, with a 40% remission rate observed in groups undergoing 6 weeks of therapy consisting of resistance training and aerobic exercise, respectively [28]. O’Sullivan et al., on the other hand, investigated the efficacy of RET in alleviating depressive symptoms among young adults presenting symptoms of GAD, often concurrent with major depressive disorder (MDD). The study revealed notable reductions in depressive symptoms following RET intervention, even among participants grappling with both anxiety and depression. Remarkably, nearly all individuals meeting MDD criteria experienced remission by the eighth week of the study. Additionally, RET demonstrated promising antidepressant effects among individuals experiencing clinically significant anxiety, suggesting its potential as either a primary or adjunctive treatment for moderate depressive symptoms within this cohort. The study also proposed a synergistic relationship between the reduction of anxiety and depression symptoms, possibly attributed to shared underlying causes and the heightened benefits of RET for those burdened with severe anxiety [29]. The implementation of physical exercise in patients with GAD and Heart Failure (HF) has shown promising results. Participation in a community exercise program, which included weekly sessions of 1 h consisting of aerobic training, resistance training, and flexibility/balance exercises over 12 weeks, was associated with significant reductions in patient health questionnaire (PHQ) somatic symptoms. Furthermore, this physical activity intervention demonstrated a significant anxiolytic effect. Notably, the current HF self-management program (HFSMP) was only government-funded for one session weekly. This limitation suggests that increasing the frequency of exercise sessions could potentially enhance the observed benefits [30]. 

Both RET and aerobic exercise training (AET) improved symptoms associated with GAD. The authors indicate that short-term exercise training, including resistance exercise, can effectively alleviate symptoms related to GAD, particularly irritability, anxiety, low vigor, and pain [31]. Herring et al. directed their attention toward evaluating the quality of sleep after short-term exercise interventions. They observed enhancements in sleep initiation, continuity, and reductions in time spent in bed and hypersomnia among young women diagnosed with GAD. The RET protocol involved progressively intensifying leg press, leg curl, and leg extension exercises over seven sets of 10 repetitions. Additionally, the AET regimen comprised two weekly sessions of 16 min of continuous leg cycling, designed to parallel RET in workload and intensity progression [32].

#### 3.1.2. Yoga

Researchers found that yoga was more effective in reducing anxiety symptoms in patients with GAD than stress-related education. The incorporation of yoga breathing practices may be effective in managing temporary stress and enhancing overall mental health. However, researchers noted that yoga was less effective than CBT [17]. The study conducted by Szuhany et al. [33] indicates that while preferences for CBT and yoga are similar among patients with GAD, treatment outcomes are not necessarily improved by aligning treatments with patient preferences. Notably, there is a significant difference in response rates for CBT (82%) compared to yoga (45.2%) when the treatment does not match the patient’s preference. Furthermore, the study observed a higher dropout rate among participants matched to their yoga preference (63%) compared to those not matched, a trend not seen with CBT (40%).

The authors suggest that these findings may be influenced by the predominantly White and well-educated sample, which may limit the generalizability of the results to more diverse populations. Additionally, the specific type of yoga used in the study (kundalini yoga) may not represent the effects of other yoga styles, further affecting the generalizability of the findings [33]. The findings from the present study align with those of Simon et al., demonstrating that both interventions, namely yoga and CBT, surpassed stress education in terms of efficacy. Specifically, the response rates were noteworthy: 54.2% in the yoga group and 70.8% in the CBT group, contrasting with a mere 33% in the stress education cohort. However, despite these promising outcomes, the equivalence analysis failed to establish Kundalini yoga’s parity with CBT in effectiveness. These results underscore the potential of Kundalini yoga as a viable therapeutic option for GAD. Nevertheless, CBT retains its status as the primary treatment modality for this condition [34]. Doria et al. evaluated the effectiveness of the Sudarshan Kriya Yoga (SKY) protocol in treating GAD and depression among patients on pharmacological treatment and those not using psychiatric medication. Both groups participated in a two-week intensive SKY segment, involving rhythmic breathing with closed eyes and focused awareness. Significant improvements in anxiety and depression scores were observed in both groups, with further enhancements noted over six months of weekly follow-ups. The lack of significant differences between the groups suggests that SKY is an effective complementary therapy for those on medication and a viable primary treatment for those not using medication. Additionally, there was a marked reduction in Symptom Checklist-90 Global Severity Index (GSI) scores, linked to increased self-awareness and self-efficacy from regular SKY practice. Patients reported improved self-esteem and confidence in managing crisis symptoms, leading to reduced demands on the healthcare system. This optimization of public health costs supports SKY as a reliable adjunct or alternative treatment, particularly for those with poor response to or inadequate adherence to pharmacological treatment [35]. 

### 3.2. Social Anxiety Disorder

#### 3.2.1. Resistance and Aerobic Exercise

In patients diagnosed with SAD, a reduction in anxiety and depressive symptoms resulted from interventions involving mindfulness-based stress reduction (MBSR) and aerobic exercise. These effects were immediately post-intervention and maintained at follow-up after 3 months [36]. Goldin et al. [37] investigated the impact of AE and MBSR on self-reported negative emotional reactivity and brain activity in patients with SAD. Both interventions were effective in reducing self-reported negative emotional reactivity to negative self-beliefs (NSBs). However, when comparing reactions to NSBs versus reading neutral statements, distinct effects on brain activity emerged.

MBSR was associated with a decrease in signal in the right ventrolateral prefrontal cortex (PFC), indicating reduced automatic cognitive control of emotional experiences during NSB reactions. This reduction aligns with MBSR’s emphasis on fostering flexible attentional engagement with negative self-beliefs. Moreover, MBSR led to increased responses in parietal brain regions associated with attentional alerting, suggesting heightened attentional engagement with emotional stimuli.

On the other hand, AE was linked to an increased signal in the right ventrolateral PFC, possibly reflecting enhanced cognitive control processes. However, AE was also associated with decreased responses in parietal brain regions, indicating reduced attentional engagement with emotional stimuli compared to MBSR. Additionally, MBSR participants who engaged in more meditation practice exhibited increased responses in brain regions associated with cognitive and visual attention, a correlation not observed with AE practice [37].

Interestingly, Jazarei et al. found that those with lower pre-treatment social anxiety experienced significantly fewer weekly social anxiety symptoms while participating in MBSR compared to those undergoing aerobic exercise. Conversely, participants with higher pre-treatment social anxiety in the MBSR group showed slightly elevated levels of weekly social anxiety symptoms compared to their counterparts in the AE group. However, both MBSR and AE resulted in significant and comparable reductions in weekly social anxiety symptom trajectories, indicating that there were no significant differences between the effectiveness of the two interventions [38].

#### 3.2.2. Yoga

Javed et al. collected pre- and post-data based on a daily 30-minute yoga routine. Follow-up visits occurred every 15 days to assess progress. The patient showed progressive improvement in SAD scores, with reductions in the severity of issues such as bladder shyness and agoraphobia [39].

Investigating the efficacy of yoga in treating individuals with SAD presents several complexities. This disorder encompasses anxieties linked to diverse social scenarios, complicating the generalization of findings. Moreover, the multitude of yoga types contributes to varying intervention effectiveness. While movement-based approaches might effectively alleviate overall anxiety, breath-focused methods could be more advantageous in specific situational exposures. Additionally, factors like seasonal changes, lifestyle modifications, and stress levels may impact study outcomes longitudinally. Hence, concluding yoga’s efficacy in treating SAD requires careful consideration of the disorder’s intricacies and the diverse range of interventions available. Despite these limitations, yoga, particularly hatha yoga, has shown promise in alleviating symptoms of anxiety and depression, particularly in non-clinical populations or those with subthreshold symptoms. Therefore, while promising, the overall applicability and effectiveness of yoga for anxiety disorders remain inconclusive [40].

### 3.3. Post-Traumatic Stress Disorder

#### 3.3.1. Resistance and Aerobic Exercise

High-intensity resistance exercise effectively reduced both anxiety and depressive symptoms in patients with PTSD, with effects observed within a 3-week study period [41]. Regular supervised aerobic exercise sessions, conducted over a two-week period on a stationary bicycle at 60–80% of the heart rate reserve, have been shown to significantly reduce symptoms of PTSD in individuals. Furthermore, sustained benefits were observed at a one-month follow-up. These findings highlight the potential of aerobic exercise as a complementary approach to traditional psychotherapy for managing PTSD [42]. Rosenbaum et al. validated the findings outlined above in their research, demonstrating that an exercise intervention consisting of resistance training and a pedometer-based walking program significantly improved PTSD symptoms. Participants who underwent this intervention reported a significant decrease in PTSD symptoms compared to those who received standard care, including psychotherapy, pharmaceutical interventions, and group therapy. Additionally, participants engaging in the physical intervention also saw reductions in depressive symptoms, waist circumference, and improvements in sleep quality [43].

#### 3.3.2. Yoga

For individuals with PTSD, yoga therapy led to a reduction in anxiety and depressive symptoms both immediately after the intervention and at a follow-up after 3 months, compared to a control group. However, it is essential to note that nonpsychoactive medications were allowed if the dosage had been stable for at least one month. Additionally, researchers observed decreased intrusion and avoidance symptoms following the intervention [44]. Carter et al. arrived at similar conclusions, noting significant improvements in PTSD scores at 6 weeks and further enhancements at the 6-month follow-up following a 5-day multi-component yoga course tailored for veterans. The authors emphasized the apparent effects despite these individuals having a 30-year history of treatment-resistant severe PTSD, alcohol abuse, and reliance on disability benefits. They suggest that yoga could feasibly serve as a preventive intervention for individuals at high risk of developing PTSD [45]. However, it is noteworthy that despite the positive psychological effects of the yoga intervention in PTSD patients, researchers did not observe any changes in salivary cortisol levels or inflammatory markers. These findings suggest that while yoga provided psychological benefits, it did not significantly affect these physiological aspects [46]. Yoga offers a potentially holistic approach to coping with PTSD symptoms, facilitating the management of physiological and sensory responses associated with fear and vulnerability, as well as increasing emotional awareness and resilience. The authors report that yoga has shown promise in reducing PTSD symptoms, producing effects similar to those observed with established psychotherapeutic and psychopharmacological interventions [47]. Davis et al. conducted a study involving 209 participants, mostly veterans, who had been diagnosed with PTSD. They were randomly assigned to either a yoga program (HYP) or a walking and teaching program (WLP), attending sessions twice a week for 16 weeks. Results showed that the yoga program significantly reduced the severity of PTSD compared to WLP at the end of treatment. However, these differences did not persist at the 7-month follow-up. Yoga may be a beneficial complementary intervention for PTSD alongside conventional treatments. Future studies should explore the addition of social components or booster sessions for long-term benefits [48]. In a randomized controlled trial focusing on women with PTSD, a population with limited literature available, yoga intervention did not lead to significant changes in physical activity or self-efficacy. However, there was a notable decrease in external motivation. Interestingly, the control group experienced decreased amotivation scores during the study. The authors speculate that this may be attributed to participants planning to join yoga classes after data collection, potentially elevating their motivation levels in anticipation of regular exercise. Participants engaged in a 75-minute yoga session weekly for 12 weeks, or twice weekly for 6 weeks. The intervention incorporated trauma-sensitive elements, gradual progression of simple poses, and components of dialectical behavior therapy [49]. 

In another randomized control feasibility trial, Huberty et al. demonstrated that yoga interventions in women who have experienced stillbirth resulted in short-term reductions in PTSD and depressive symptoms. However, uncertainties persist regarding the long-term effects, indicating the necessity for further research. Remarkably, even modest weekly yoga durations, such as 77 min, showed potential for improving depressive symptoms and perinatal grief. Although previous studies support similar outcomes with 60–75 min of yoga per week, it is crucial to acknowledge the predominant use of in-person interventions compared to the home-based format in this study. Additionally, while yoga displays promise in reducing depressive symptoms and grief, these emotional states may serve as barriers to participation, potentially affecting engagement levels. Furthermore, although the dose-response analysis yielded non-significant results, trends suggest that increasing weekly yoga minutes may lead to more pronounced improvements in post-traumatic stress and depression scores. Therefore, while even minimal yoga durations may initiate positive changes, more weekly minutes may provide enhanced benefits, emphasizing the significance of considering feasibility factors in interpreting these findings [50]. 

### 3.4. Panic Disorder

#### 3.4.1. Resistance and Aerobic Exercise

According to Ströhle et al., engaging in 30 min of aerobic exercise can effectively alleviate acute panic attacks and the accompanying anxiety symptoms. Furthermore, exercise decreased the likelihood of individuals experiencing panic attacks induced by cholecystokinin tetrapeptide (CCK-4) [51]. Another study combining CBT with aerobic exercise, specifically running on a treadmill to achieve 70% of patients’ maximal oxygen consumption (VO_2_ max), yielded a superior reduction in anxiety symptoms compared to CBT alone. Moreover, the combination therapy exhibited a sustained and increasing therapeutic effect over time, lasting from 3 to 7 months [52]. A study involving patients with PD and agoraphobia revealed that combined therapy with paroxetine and either aerobic exercise or placebo led to a reduction in panic anxiety and depressive symptoms across both intervention groups. However, the group receiving paroxetine demonstrated a significant anxiety reduction compared to the group not receiving the drug. These findings suggest that aerobic exercise may serve as an effective adjunct to conventional therapy for PD and agoraphobia [15]. Similarly, Broocks et al. found that both regular aerobic exercise over a 10-week and clomipramine treatment at 112.5 mg/day significantly improved symptoms in individuals with moderate to severe panic disorder compared to placebo. While exercise alone was less effective than clomipramine, its role in treatment remains valuable, particularly for patients unable or unwilling to take medication. Integrating exercise into treatment plans could offer meaningful benefits, especially considering the lack of contraindications for aerobic exercise in many individuals with panic disorder and agoraphobia, who are typically younger [53].

#### 3.4.2. Yoga

Ensari et al. studied 18 participants engaging in guided yoga and light stretching sessions, combined with a 7.5% CO_2_ inhalation task. They assessed anxiety and panic states before and after each task while monitoring respiratory measurements during inhalation. They observed a slight reduction in cognitive anxiety post-activity, hinting at a potential overall effect of physical activity in mitigating cognitive anxiety, irrespective of physiological changes. These findings suggest the potential utility of yoga as a tool for reducing perceived anxiety [54]. The investigation among individuals diagnosed with PD unveiled notable reductions in anxiety levels, panic-related beliefs, and sensations associated with panic. Significant enhancements were observed in the yoga-only and the combined CBT and yoga group. Nevertheless, the synergy of CBT and yoga yielded superior outcomes compared to yoga alone. In a group with only yoga, participants attended classes twice a week, for 50 min each session, over a period of 2 months. In the group with CBT and yoga, participants practiced yoga once a week for 50 min, CBT once a week for 50 min, over 2 months [55]. Consistent with these findings, another randomized controlled trial demonstrated that the yoga group experienced a significant reduction in anxiety scores compared to the control group, which could receive standard care, including pharmacotherapy and cognitive-behavioral therapy. Additionally, by the 12th week, improvements were observed across all domains of quality of life within the yoga group, highlighting yoga’s comprehensive impact on well-being. The study utilized a standardized yoga program, specifying the practices included in the intervention, such as asana, pranayama, and meditation. However, the study’s limitation lies in its participant pool, as all individuals were diagnosed with panic disorder and recruited from a single location, potentially affecting the generalizability of the findings [56].

### 3.5. Conclusions

In conclusion, various forms of physical activity, including resistance training, aerobic exercise, and yoga, have shown significant efficacy in reducing anxiety and depressive symptoms in different anxiety disorders such as GAD, SAD, PTSD, and PD. Integrating these forms of activity into standard therapeutic programs can provide substantial patient benefits.

Physical exercise, including yoga, appears to hold potential as an adjunct to drug therapies and CBT for anxiety disorders. Conversely, drug-based therapy and physical exercise, which exert their effects over time, may prove more effective in the long term [16]. Moreover, research suggests that physical activity may play a role in delaying or preventing the onset and progression of GAD and AGAD.

Incorporating psychotherapeutic interventions into physical exercise can enhance treatment effectiveness. Cognitive therapy (CT), which targets and restructures distorted thought patterns contributing to anxiety, is particularly effective for conditions like GAD and SAD [57]. CT, when combined with exercise, can reinforce positive cognitive changes, improving overall outcomes. For instance, regular aerobic exercise reduces anxiety symptoms, and when paired with CT, it can lead to more significant cognitive shifts that support long-term recovery [58].

Integrating physical exercise with behavioral therapy (BT) can have potential therapeutic effects in the treatment of panic disorder and specific phobias, where avoidance significantly impairs daily functioning. For example, aerobic exercise, as a form of exposure therapy, allows patients to confront their fears in a controlled environment while taking advantage of the anxiolytic properties of physical activity [59]. 

Early intervention is crucial, as it significantly reduces the risk of developing future psychopathological disorders. The absence of contraindications for discontinuing physical activity in psychiatric patients without additional medical conditions renders it a cost-effective tool with significant potential for reducing anxiety symptoms. However, it is essential to note that using these methods alone, particularly in patients with severe mental conditions, is not advisable. Most studies have examined these interventions as adjuncts to traditional therapy rather than standalone treatments. Gradually increasing training intensity, with the guidance of an experienced trainer/therapist, particularly during the initial session, is crucial for achieving optimal results [60]. The ability to adjust the intensity of exertion, ranging from low to intense, allows for personalized recommendations based on the level of anxiety experienced. Higher exertion levels offer immediate relief from acute stress, whereas lower levels can be employed over the long term to alleviate daily mental tension. Integrating yoga with conventional therapies such as CBT can yield synergistic effects and long-term benefits in reducing anxiety symptoms, especially for those experiencing panic disorder.

It is essential to acknowledge the limitations of existing research, particularly regarding the demographic characteristics of sample populations. Many studies have predominantly involved White, well-educated participants, which may limit the generalizability of findings to more diverse populations. Future research should aim to include a broader range of demographic backgrounds, including varying socioeconomic statuses, ethnicities, and educational levels, to understand better how these interventions can be adapted to meet the needs of all individuals suffering from anxiety disorders. Additionally, exploring the effects of physical activity in underrepresented groups can provide valuable insights into the efficacy of these interventions across different populations, ultimately enhancing treatment strategies for diverse patient demographics.

It is worth noting that in some studies involving yoga, particularly with PTSD patients, there have been high dropout rates. These may be due to physical challenges or the severity of anxiety symptoms. To reduce dropout among patients with anxiety disorders, consider offering yoga classes at local psychological support centers or providing them online. This approach can create a supportive environment and encourage consistent participation.

Conducting a well-designed study on the effects of exercise on anxiety disorders presents significant challenges. The subjective nature of measuring anxiety severity and the varying physical conditions of individuals make it difficult to draw generalized conclusions for a broad patient population.

Determining appropriate training volumes for individuals adds further complexity, as each person may require different exercise intensities, complicating the creation of uniform research protocols. Moreover, ensuring controlled study conditions is challenging when patients exercise at home, where adequate monitoring of their sessions is impractical.

To address these methodological challenges, future research should focus on exploring the therapeutic effects of different exercise modalities across various anxiety disorders. Investigating the mechanisms behind the anxiolytic effects of exercise and customizing interventions to meet individual needs can enhance the effectiveness of exercise-based treatments. By addressing these complexities and individual variabilities, researchers can improve the understanding of how exercise impacts anxiety disorders and optimize treatment strategies for diverse patient populations. 

## 4. Virtual Reality (VR) 

Virtual reality is a technology that utilizes computer-generated simulations to replicate experiences resembling authentic environments in terms of visual, auditory, and sensory aspects [61]. VR enables users to immerse themselves in computer-generated environments, facilitating natural interaction with virtual functions or objects through a human-computer interface [62]. The components of this interface that mediate in creating the VR environment include goggles for displaying images, headphones, controllers that track the user’s position, and a computer or console required for generating graphics (Figure 2).

A fundamental element contributing to the potential widespread implementation of VR in anxiety disorder therapy is the adaptability of the environment and the capacity for repetitive exposure to the situations it presents [63]. Researchers conducted a study on patients with various mental disorders to compare the effectiveness of relaxation in virtual reality (VRelax) with traditional relaxation exercises over 10 days. The study found that VRelax immediately reduced negative feelings, especially feelings of anxiety and depression. The effect was more pronounced in patients who were simultaneously undergoing outpatient psychiatric treatment, suggesting the potential use of VR as a supplement to pharmacological therapy [64]. 

### 4.1. Post-Traumatic Stress Disorder

An intriguing therapeutic option emerges as gradual exposure therapy in virtual reality (VR-GET), which has demonstrated a notable reduction in subjective feelings of anxiety when employed in veterans with PTSD. Concurrently, clinicians observed a slight, clinically insignificant impact on reducing depressive symptoms. The treatment regimen comprised weekly or bi-weekly 90-minute sessions with a psychologist, and the VR-GET therapy spanned 10 weeks [65]. These findings align with the results of Beidel et al., who suggest that virtual reality exposure therapy (VRET) effectively diminishes symptoms associated with PTSD in soldiers. Importantly, they observed that the beneficial effects persisted during the 3 and 6-month follow-ups, with a low recurrence rate of 4.5% [66]. The study conducted by Katz et al. suggests that both virtual reality exposure therapy (VRE) and exposure therapy (PE) effectively diminish skin conductance reactivity (GSR) in response to traumatic experiences. It is important to highlight that the group receiving VRE therapy demonstrated more significant clinical improvements than the control group [67]. Of note, soldiers who achieved more substantial improvements in VRE were characterized by younger age, absence of antidepressant medication, higher levels of excessive anxiety symptoms, and risk of suicide higher than minimal. The authors highlight the necessity of conducting research involving a more diverse gender group of patients, emphasizing that the study results should not generalize to the entire population of individuals with military-related PTSD. 

### 4.2. Social Anxiety Disorder

VRE has shown efficacy in treating social anxieties, with the benefits sustained for up to one year. Importantly, this therapy is comparable in effectiveness to exposure group therapy (EGT). The authors acknowledge certain limitations in the study, including its exclusive focus on fear of public speaking, which restricts its applicability to a broader spectrum of social situations that individuals with SAD may encounter [68]. Anderson et al. suggest that VR can serve as an effective tool in CBT aimed at alleviating anxiety associated with fear of public speaking (FOPS). The authors observed significant improvements in self-esteem and reductions in public speaking-related anxiety among patients, immediately after therapy and during a 3-month follow-up period [69]. A similar study observed that VR stimulation, wherein patients were exposed to avatars’ faces, correlated with increased activity in the visual cortex. This suggests a decreased tendency to avoid eye contact among patients with SAD [70]. The authors have observed increased activity in the medial dorsal nucleus of the thalamus, responsible for fear extinction modulation, following VRS therapy. They propose utilizing the study of this region to assess enhancements in regulating emotional signals and forecasting therapy outcomes [71]. Another potential approach to address visual avoidance during public speaking among individuals with SAD involves VRE combined with Attentional Focus Training (AGT). The authors hypothesize that this combination leads to a reduction in public speaking anxiety and general symptoms of social anxiety [72].

A study examining the efficacy of VRE as a treatment modality for individuals with SAD and various interpersonal fears offered patients the choice of scenario: an interview or an informal dinner with strangers. The findings revealed that VRE therapy resulted in more significant reductions in SAD severity, job interview anxiety, and worry compared to the waiting list (WL) control group. The sustained improvements were observed during 3- and 6-month follow-up assessments [73].

Bouchard et al. argued that CBT with in virtuo exposure is more effective and practical for therapists treating individuals with SAD than CBT with in vivo exposure. Furthermore, they note that the benefits of exposure therapy in virtual reality persisted up to a 6-month follow-up period [74]. Kampmann et al. showed that VRET and individual in vivo exposure therapy (iVET) groups experienced improvements in social anxiety symptoms, speech duration, perceived stress, and beliefs related to avoidant personality disorder compared to the waiting-list group. However, only the iVET group showed additional benefits in reducing fear of negative evaluation, enhancing speech performance, and lowering general anxiety, depression, and improving quality of life. At follow-up, iVET participants maintained significant improvements in all measures, while VRET participants only showed reduced perceived stress. These findings suggest that in vivo exposure therapy has advantages over virtual reality exposure therapy, providing more comprehensive and sustained benefits for individuals with social anxiety disorder [75]. In one study using virtual reality (VR) for relaxation, the effects were more pronounced at the beginning of the study than at the end. This diminished effect may be due to the limited selection of videos available for self-play, which could have become monotonous over time. Notably, the video featuring diving with dolphins was the most popular among participants, likely because it required more focused attention to follow the movement of the dolphins. That suggests the potential benefit of exploring different types of virtual scenery to maximize relaxation effects [64].

### 4.3. Generalized Anxiety Disorder

In contrast, GAD patients exhibited increased alpha electroencephalography (EEG) activity after a single 20-min bike ride in VR, implying that exercising in an immersive virtual environment may induce greater feelings of relaxation in these patients. Notably, utilizing a natural image environment (VN) proved more effective than environments containing abstract images (VAPs). It suggests the potential for using VR exercise to alleviate anxiety symptoms in patients who are hesitant about interacting with the environment outside their homes. However, the authors underscore the necessity of researching the long-term effects of VR exercises, as the prior study only addresses a single session [76]. In the study conducted by Corini et al., participants underwent an eight-session VR-based therapy that included relaxation and exposure techniques, augmented by HR biofeedback and supplementary home training via a cell phone. The use of virtual reality in treating generalized anxiety disorder (GAD) demonstrated significant efficacy, particularly when combined with mobile phone enhancements for home-based VR experiences. Notably, the inclusion of biofeedback in the Mobile Phone with Biofeedback Condition (VRMB) group resulted in substantial reductions in anxiety scores post-treatment. These findings indicate that integrating VR therapy with mobile biofeedback technology can markedly improve treatment outcomes for GAD [77].

### 4.4. Phobias

Pitti et al. assert that therapies combining paroxetine with CBT, as well as paroxetine with CBT and virtual reality (VR) exposure, proved more effective in treating agoraphobia compared to paroxetine alone. Both models involved 11 CBT sessions, supplemented by 4 exposure sessions in the VR group over three months [78]. In contrast, the combination of virtual reality exposure therapy (VRET) and CBT did not yield a superior therapeutic effect compared to VRET alone in patients with agoraphobia. Following 10 weeks of VRET therapy, a reduction in anxiety and avoidance, along with an improvement in general well-being, was observed. There is an opportunity to utilize VRET as an alternative to CBT rather than an adjunct. Side effects of using VR include cybersickness, characterized by symptoms such as nausea, excessive sweating, and dizziness [79]. A study by Tortella-Feliu et al. compared three computer-assisted exposure therapies: VRET, computer-assisted exposure with a therapist (CAE-T), and computer-assisted self-exposure (CAE-SA). The results indicated that all three methods were equally effective in reducing fear of flying, with no significant differences in efficacy. Moreover, CAE-SA did not exhibit a higher dropout rate, demonstrating good acceptability and adherence. These findings suggest the potential to minimize therapist involvement in computer-based therapies without compromising their effectiveness [77]. VRET demonstrates promising results in reducing driving anxiety and avoidance behaviors in individuals with driving phobia. VRET effectively improves phobia-related symptoms, decreases peak anxiety levels, and eliminates diagnostic criteria for driving phobia in some patients. Although VRET may not suffice for all individuals, it provides a valuable initial treatment step, potentially followed by in vivo exposure therapy as needed [80]. A study by Powers et al. highlighted that, in virtual avatar therapy, individuals with social and generalized anxiety reported heightened levels of fear after interacting with an avatar in VR compared to conversing with a therapist in person [81]. 

### 4.5. Conclusions

The utilization of virtual reality (VR) for exposure therapy emerges as a promising option for patients grappling with anxiety disorders associated with specific situations, notably PTSD, specific phobias, or SAD. Furthermore, VR therapy presents a viable alternative for patients experiencing anxiety when leaving their homes, thereby impeding therapeutic progress. In such cases, attention should be directed towards the quality of the graphics offered during therapy. Regrettably, the superior quality of the technology employed in these studies often comes with increased therapy costs, posing a potential limitation for many patients. Therefore, there is a pressing need for further research encompassing various scenarios of anxiety-inducing situations in patients with SAD or PTSD. An interesting approach seems to be combining VR therapy with insight-oriented psychotherapy, which aims to encourage patients to explore their thoughts, feelings, and experiences to gain insight into their anxiety. By understanding the root causes of their distress, patients can develop healthier coping mechanisms and emotional regulation strategies [82]. IOP has therapeutic potential, especially in the group with PTSD, where past trauma significantly influences current symptoms. Integrating VR into IOP can provide a safe space for patients to confront traumatic memories and practice new coping strategies, enhancing the therapeutic experience. However, there is a need for further exploration of psychotherapeutic therapies in combination with VR to clarify conclusions as to which will be most beneficial. Many of the conducted studies shared common limitations, such as small control groups or subjective assessments of symptom severity associated with anxiety disorders before study enrollment.

## 5. Mindfulness

Mindfulness-based interventions (MBIs) encompass mindfulness-based stress reduction (MBSR) and mindfulness-based cognitive therapy (MBCT). MBSR, developed by Jon Kabat-Zinn in the 1970s, aims to alleviate stress by fostering mindful awareness of the present moment. Initially designed for stress management and relaxation among individuals coping with chronic pain, the standard MBSR program consists of 2-hour sessions held once a week for 10 weeks [83]. On the other hand, MBCT integrates CBT principles with MBSR techniques. The original format entails 2-hour sessions once a week over 8 weeks, typically conducted in groups of up to 12 participants [84]. Due to its efficacy in reducing the frequency of recurrent depressive episodes, the British National Institute for Health and Care Excellence (NICE) has incorporated it into the treatment recommendations as a viable option for individuals with recurrent depression [85]. While MBSR and MBCT emphasize mindfulness, MBSR is more general in its application to stress and physical health. In contrast, MBCT is specifically tailored to address cognitive patterns associated with depression. This distinction highlights the unique therapeutic approaches of each program, making them suitable for different populations and mental health challenges. 

One of the fundamental mechanisms of action of MBIs is the reduction of ruminations and worries, which are common cognitive patterns associated with anxiety and depression. Studies have shown that MBIs can significantly reduce these repetitive thought processes, thereby alleviating symptoms of anxiety [86]. In addition, MBIs promote self-compassion and emotional regulation, enabling individuals to respond more adaptively to stressors and negative emotions [87]. Furthermore, MBIs increase cognitive flexibility and attentional control, allowing individuals to manage their emotional responses better and improve problem-solving skills [88]. This is particularly important in populations at risk for suicidal behavior, where improved emotional regulation can mitigate impulsive actions. Overall, integrating mindfulness practices fosters greater awareness of internal experiences, leading to healthier coping strategies and improved mental resilience in various populations, including adolescents and the elderly [89].

### 5.1. MBI and Classical Treatment

In patients with anxiety disorders who had previously undergone CBT therapy, the use of MBCT led to a more substantial reduction in subjective anxiety at 2 months compared to the group receiving cognitive behavioral therapy for relapse prevention (CBT-RP). However, follow-up after 6 months indicated that the effect of MBCT on anxiety severity was not sustained in the long term. It could be attributed to the fact that these patients had already experienced ineffective psychotherapeutic treatments, and maintaining long-term effects in such cases requires additional booster sessions [90]. Conversely, in patients with agoraphobia or PD who had no prior history of CBT, MBCT showed symptom improvement at the 2-month follow-up. The authors suggest that MBCT therapy offers rapid action and long-term benefits. However, the limitations include the size of the sample and the absence of a control group [91]. Li et al. propose that MBI and CBT exhibit similar effectiveness in alleviating symptoms of anxiety, depression, and sleep problems [92]. These findings align with another study, where MBCT demonstrated a comparable reduction in anxiety symptoms compared to CBT. MBSR intervention exhibited lower efficacy than MBCT therapy. The difference may arise from the individualized approach of MBSR in addressing anxiety, while MBCT integrates cognitive-behavioral therapy and involves supervision by a trained therapist. The authors emphasize the lack of sufficient high-quality research on the long-term effects of MBI interventions compared to CBT [93].

### 5.2. Social Anxiety Disorder

Koszycki et al. observe a significant reduction in social anxiety among SAD patients undergoing group therapy (CBT) compared to MBSR. Both interventions effectively enhance mood and quality of life. The authors propose considering MBSR as an alternative treatment option for patients with SAD [94]. In contrast, Goldin et al. find no notable differences in the anxiolytic effects between CBT and MBSR [95]. In a separate study, the spontaneous use of MBSR following exposure to an anxiety-inducing situation showed heightened neural responses in brain regions associated with visual attention on fMRI. Experts believe this is associated with nostril movement during breathing or heightened visual engagement while focusing on the breath. Patients practicing breath-focused MBSR demonstrated superior emotional reactivity control compared to attention-redirecting methods. The study’s limitation includes the absence of a control group. Furthermore, the examiner created the set of negative stimuli used in the study, potentially leading to weaker neural behavioral responses [96]. Both cognitive restructuring and mindfulness strategies proved effective in reducing post-event processing (PEP), which refers to the detailed review of one’s performance after a social situation, often negatively, serving as a significant maintenance factor of SAD. Furthermore, they were effective in enhancing affect compared to the control condition. Participants in these interventions reported decreased PEP, improved state affect, and more substantial reductions in skin conductance level (SCL) compared to those in the control condition. Changes in cognitive processes predicted reduced PEP at follow-up [97]. The mindfulness-based intervention for social anxiety disorder (MBI-SAD) demonstrated significant improvements in social anxiety symptom severity, depression, and social adjustment compared to the waitlist control group. The intervention comprised 12 weekly group sessions, each lasting 2 h and led by a psychologist. These sessions included psychoeducation, formal mindful meditations, mindful exposure, and compassion-based meditations. Additionally, participants showed increased levels of self-compassion and various aspects of mindfulness, with these positive effects persisting at the 3-month follow-up assessment [98]. Another study by Koszycki et al. suggests that CBGT is more effective than MBI-SAD in reducing social anxiety severity. Still, both interventions have comparable effects on other measures of well-being. This difference in effectiveness could be attributed to the distinct approaches to exposure between CBGT and MBI-SAD. In CBGT, participants actively engage in altering negative thoughts during challenging situations, potentially leading to more pronounced anxiety reduction compared to the mindfulness skills taught in MBI-SAD. Although introducing mindfulness exposure earlier in MBI-SAD might impact outcomes, the decision to sequence the exercises was based on the importance of participants understanding mindfulness before applying it to challenging social scenarios. The trial lacked a nonspecific control condition, making it unclear whether the study interventions produced effects beyond nonspecific factors contributing to therapeutic response. These factors include therapist attention, therapeutic alliance, expectancy for improvement, and group participation and cohesion [99].

### 5.3. Generalized Anxiety Disorder

Hoge et al. observed that mindfulness training reduced HPA axis hormones and inflammatory markers in patients with GAD [100]. Similarly, a study conducted by Si-si Jiang et al. showed a more pronounced reduction in anxiety following standard MBSR therapy compared to CBT in patients with GAD. However, after a 3-month follow-up, anxiety levels in both groups were similar, suggesting that while MBSR may be more effective initially in alleviating anxiety in GAD patients, its long-term efficacy does not significantly differ from CBT therapy [101]. A study comparing the number of work absenteeism days and healthcare utilization frequency among individuals with GAD before and after participating in either MBSR or attention control classes revealed that the standard MBSR therapy group experienced fewer days of work absenteeism. Moreover, in the 24 weeks following the conclusion of MBSR therapy, a significantly reduced risk of job loss and fewer visits to mental health professionals were observed [102]. In another study, the authors suggest that there are no clinically significant differences between the MBCT and GAD anxiety reduction groups. However, psychoeducation prevailed in reducing depressive symptoms, which the authors attribute to higher attendance in this group. Cultural conflict arising from MBCT’s foundation in meditation, which may be perceived as a religious practice, could potentially contribute to lower engagement with this form of therapy [103]. A study involving patients diagnosed with panic disorder revealed that therapeutic benefits persisted for up to one year following MBCT intervention. However, it is important to note that medication use, which was not an exclusion criterion, could have strongly influenced these effects. Anxiety reduction was observed as early as the second week of MBCT, with a gradual decrease throughout the 8-week therapy duration.

Furthermore, a correlation was found between disease duration and anxiety severity assessment at 8 weeks, indicating that longer disease duration was associated with greater anxiety levels. These findings suggest that a longer-term MBCT therapy is required to achieve a clinically meaningful reduction in symptoms of panic disorder. However, it is important to note that the study lacks a control group and has a small sample size [104].

### 5.4. Conclusions

MBI interventions demonstrate potential in treating various anxiety disorders, with MBCT and MBSR serving as adjuncts or alternatives to standard therapies. According to Liu et al., there is a dose-response correlation, where the dose corresponds to the length and frequency of sessions. This underscores the ability to adjust the intensity of MBI therapy to enhance clinical outcomes for patients [105]. Mindfulness interventions entail minimal adverse events compared to pharmacological treatments, which is particularly important for patients who are intolerant to standard therapies. Furthermore, the shortage of psychiatric specialists highlights MBSR as a feasible adjunct intervention since it does not require specialized personnel for practice. Therefore, compared to conventional treatments, MBIs offer lower financial costs. MBIs seem to provide more potential for patients with GAD, possibly due to variations in the origins of anxiety and the corresponding responses seen in patients with SAD. It is crucial to underscore the scarcity of studies with adequately large sample sizes and the frequent absence of control groups.

## 6. Transcranial Magnetic Stimulation

In 2008, the FDA approved rTMS for the treatment of major depressive disorder (MDD) [106] and in 2018 for obsessive-compulsive disorder (OCD). Efforts are underway to utilize Non-Invasive Brain Stimulation (NIBS), including TMS, rTMS, and tDCS, for anxiety disorder treatment [107]. Transcranial magnetic stimulation (TMS) operates on Faraday’s principle of electromagnetic induction, generating rapidly alternating electrical currents in a circular coil positioned on the surface of the skull. This magnetic field is then transformed into a secondary electric current, capable of depolarizing neurons and triggering an action potential [108]. Consequently, the propagated impulse has the potential to modulate dysfunctional circuits in neuropsychiatric conditions. We differentiate between single-pulse transcranial magnetic stimulation (TMS) and repetitive TMS (rTMS).

rTMS differs from single-pulse TMS in that the impulses are delivered repeatedly over a short period of time in sessions, aiming to produce more lasting changes in the targeted central nervous system structures, which makes it clinically useful.

rTMS encompasses various protocols, including deep TMS (dTMS) and theta burst stimulation (TBS) [109]. Furthermore, based on the type of stimulation, it is categorized into low-frequency stimulation (<1 Hz) LF-rTMS, which has an inhibitory effect, and high-frequency stimulation (>5 Hz) HF-rTMS, which leads to the excitatory impacts in the brain [110]. The authors propose that the inhibitory and excitatory effects stem from long-term synaptic potentiation (LTP) and long-term synaptic depression (LTD). dTMS is a method used to stimulate deeper brain structures by employing coils that generate a magnetic field with greater penetration capacity. Theta burst stimulation (TBS) mimics the brain’s natural theta rhythm observed in the hippocampus, involving a sequence of high-frequency stimulations (3 pulses of 50 Hz) repeated with an interstimulus interval (ISI) of 200 ms (5 Hz). TBS offers more consistent results than rTMS, owing to its standardized intensity and pulse settings. In contrast, rTMS, unlike single-pulse TMS, can modulate cortical activity beyond the stimulation period, likely due to prolonged patient exposure. This characteristic renders rTMS a promising alternative for treating neurological and psychiatric disorders [110]. The dorsolateral prefrontal cortex (DLPFC) plays a crucial role in monitoring information stored in working memory [111]. It participates in selecting sensory information and cognitive responses, influencing emotional reactivity by modifying higher-order perceptual attention systems [112].

### 6.1. Mechanism of Action

Qin et al. investigated DLPFC activity in 27 healthy volunteers exposed to acute stress. The study authors noted a reduction in DLPFC activity related to working memory. It underscores the significance of the DLPFC region in the context of stress response, making it the primary target for stimulation during TMS procedures [113]. This finding aligns with Balderston et al.’s results, where stimulation of the right DLPFC with 10 Hz rTMS (Hf-rTMS) led to increased anxiety levels in healthy volunteers. It underscores the importance of carefully diagnosing patients before TMS therapy, as stimulating the right DLPFC may induce anxiety disorders in healthy individuals [114]. It is noteworthy to consider the two theories proposed regarding the model of prefrontal asymmetry in emotion processing. The first theory posits that initial emotional processing occurs in the right hemisphere before being transferred to the left hemisphere for higher-order evaluation and control. The second theory suggests that the right hemisphere primarily processes negative emotions, while the left hemisphere processes positive emotions. Studies have shown that anger generation is associated with the left hemisphere, whereas anxiety is linked to the right hemisphere [115]. In a study where researchers attempted to boost the activity of the brain’s left hemisphere using Hf-rTMS to increase the influence of positive emotions on the perception of new stimuli, the anticipated advantages did not materialize. Interestingly, the results were consistent across both low- and high-anxiety groups. The authors propose that inhibiting the right side of the brain may yield further insights into the impact of cerebral asymmetry on regulating our responses [116].

### 6.2. Generalized Anxiety Disorder

In the first randomized trial, Diefenbach et al. showed that rTMS targeted at DLPFC 1 Hz (LF-rTMS) resulted in a higher response rate than placebo in patients with GAD. Additionally, changes in DLPFC activity measured by functional magnetic resonance imaging (fMRI) tended to correlate with anxiety symptoms and also correlated with changes in worry symptoms [117]. Furthermore, the group receiving TMS during the decision-making gambling task showed greater activity in the right DLPFC, confirming the improvement in prefrontal cortex functioning. The intensity of activation in the right DLPFC (the stimulated site) was higher after LF-rTMS stimulation, which is typical–local inhibition contributes to the activation of deeper structures. In this case, this may have an even more beneficial effect, considering the prolonged, abnormal activation of the prefrontal cortex and weaker functioning of deeper structures in response to stressful situations [118]. This is consistent with the results of Assaf et al., who, analyzing gambling task results using fMRI in individuals with GAD, pointed to abnormalities in functional connectivity between the prefrontal cortex-prefrontal cortex and PFC-amygdala, as well as improvement in FC for the dorsal anterior cingulate ACC after LFrTMS application, accompanied by improvement in worry symptoms [119]. 

The most common adverse effect in the study by Diefenbach et al. [117] was pain at the stimulation site, affecting as many as 84.6% of the individuals subjected to rTMS. It is also worth noting that the study itself was conducted on a small group of patients–along with the control group, 25 individuals, and the placebo effect contributed to the improvement in the control group, but this was not a sustained state.

This study also does not answer the question of how effective rTMS is over time. Although an earlier study by Bystritsky et al. [120] indicated that improvement persisted in some individuals for up to 6 months, they did not include a placebo trial, which does not allow for a definitive conclusion. However, an addition to this discussion could be the later assessment of the participants from the Diefenbach study, in which researchers observed that even 3 months after the end of the study, improvement measured in self-reported emotional control and associated difficulties was still visible only in the actively treated group [121]. Not only anxiety but also sleep disorders, such as insomnia, and accompanying GAD, may be susceptible to LF-rTMS treatment, as demonstrated in their study by Huang et al. [122]. This effect may stem from the fact that insomnia also involves dysfunction, among others, of the amygdala in emotion processing. On the other hand, Wang et al. decided to analyze the effect of LF-rTMS in GAD in the case of stimulating more than one center, including in the study groups receiving, in addition to DLPFC stimulation, PPC–right posterior parietal cortex stimulation. Interestingly, the group receiving dual stimulation and the highest number of pulses per session-1500 showed the greatest improvement, which persisted even after 3 months. However, dual stimulation raises questions about the safety of such a procedure. In the presented study, only one patient in the group with the highest improvement discontinued participation in the study due to headaches [123].

On the other hand, Dilkov et al. in their randomized study used HFrTMS on the right DLPFC, showing improvement on the Hamilton Anxiety Rating Scale (HARS), which was still visible 4 weeks after treatment. Furthermore, patients with GAD, before the study, had high scores on the HARS scale, despite pharmacotherapy, but rTMS helped induce improvement and reduction in the scale, indicating its usefulness as an adjuvant in GAD therapy [124]. An interesting fact is that the use of such different frequencies resulted in improvement in the condition of patients. It is possible that the use of TMS itself has a modulatory effect on cortical excitability, and stronger stimulation still achieves the same effect as inhibitory stimulation, which is also ultimately aimed at activating deeper structures involved in emotion and anxiety processing in GAD.

Taking into account the presented data, it seems that TMS is a good adjunctive method in the treatment of GAD. In studies, it is noticeable that this method is often used when pharmacological treatment alone is insufficient and adjunct therapy is needed. It seems that TMS, especially LF-rTMS over DLFPC, is an effective choice, and an increasing number of studies focusing on this protocol should allow for assistance to selected patients. It is also worth emphasizing that not every patient seems to respond to TMS in accordance with our expectations, as observed already in the first attempts to use rTMS in GAD [120], where 6 out of 10 individuals experienced significant improvement due to treatment. The mechanism of action of rTMS itself remains a mystery, especially since the results also indicate that HF-rTMS may be effective in GAD. However, it is clear that it is a relatively inexpensive method to use, and the coexistence of anxiety disorders with depression, where TMS is already commonly used, theoretically increases the chances of the method’s pleiotropic effects and reduction of symptoms of both anxiety and depression [125]. 

### 6.3. Social Anxiety Disorder

Paes et al. applied LF-rTMS over the right medial PFC and HF-rTMS over the left PFC, assuming that individuals with SAD may have a predominance of right hemisphere activity. After 4 weeks, the researchers demonstrated that rTMS could be an effective treatment method for two described patients with SAD [126]. Furthermore, Minervini et al., in their study, also indicated that in the case of SAD, DLPFC stimulation with 10 Hz rTMS may exacerbate symptoms, so LFrTMS appears to be a better therapeutic option [127]. Tomita et al., on the other hand, investigated self-focused attention in SAD, where using transcranial static magnetic field stimulation (tSMS) (a non-invasive brain stimulation method capable of temporarily suppressing brain function beneath the magnet), they inhibited the activity of the right frontopolar area (rFPA), associated with increased self-focused attention. The researchers demonstrated that this method led to the suppression of overactive rFPA in individuals with anxiety disorders, measured using functional near-infrared spectroscopy (fNIRS), and indirectly, through improvements in SFASFA (body sensations and observer perspective) and DM (detached mindfulness), supporting speech task expressions [128]. 

Regarding SAD or phobias [129], there is still relatively little research on the use of TMS or rTMS. It seems that these may be effective methods of therapeutic support, but the amount of evidence is limited. Further research is needed to establish the effectiveness of such procedures.

### 6.4. Post-Traumatic Stress Disorder

Cohen et al. found that 10 daily sessions targeting the right dorsolateral prefrontal cortex with high-frequency 10 Hz rTMS (HF-RDLPFC) resulted in more pronounced therapeutic effects compared to low-frequency 1 Hz stimulation (LF-RDLPFC) or sham stimulation in patients with PTSD [130]. Conversely, in a study conducted by Boggio et al., left-sided stimulation of the DLPFC was associated with an improvement in depressive symptoms, while right-sided stimulation alleviated anxiety. Although there was a decrease in PTSD symptoms following HF-rTMS over the left DLPFC, it was less pronounced than the improvement observed after right-sided stimulation [131]. Studies by Watts et al. and Isserles et al., involving patients with PTSD, yielded consistent results despite significant differences in the duration since diagnosis. The average time since diagnosis in the first study was 39.8 years, while in the second study, it was 15.8 years. However, both cohorts of patients demonstrated sustained benefits from TMS therapy for at least 2 months post-treatment. These findings suggest that a prolonged duration since the diagnosis of PTSD does not negate the beneficial effects of TMS therapy, which could be crucial for individuals with a chronic course of this disorder [132,133].

In contrast to prior studies, Grisaru et al. employed a slow magnetic stimulator on 10 patients diagnosed with PTSD for an average duration of 5.5 years. After administering a single session of low-frequency transcranial magnetic stimulation (TMS) at 0.3 Hz to the frontal areas on both sides of the brain, the authors observed improvements in avoidance, anxiety, and somatization. The therapeutic effects lasted for 24 h. While TMS shows promise as an effective tool for urgent therapeutic interventions, this study is subject to limitations, including the small sample size and the fact that 60% of patients were using medication for PTSD during the study [134].

### 6.5. Conclusions

The ability to customize therapy through different TMS protocols and session frequencies offers the potential for personalized treatment. High-frequency transcranial magnetic stimulation (TMS) targeting the right dorsolateral prefrontal cortex (DLPFC) has shown promising outcomes in treating post-traumatic stress disorder (PTSD). TMS has minimal side effects, commonly reported as headaches, dizziness, muscle pain localized at the stimulation site, and neck discomfort. However, several limitations exist across the referred studies, such as small sample sizes, concurrent medication usage by participants, and varying durations since disease onset, all of which could influence patient improvement prognostics. Recognizing the variety of TMS protocols utilized is crucial, as their findings may not be universally generalizable. Furthermore, during patient selection for therapy, careful consideration of accurate diagnosis is essential, as TMS usage may induce anxiety in otherwise healthy individuals.

#### Neurostimulation: New Directions

Recently, there has been a perspective on the use of a new, promising method-transcranial near-infrared stimulation (tNIRS). This is a non-invasive infrared stimulation, with an optimal wavelength of about 820 nm (due to the ability to penetrate the skull and absorption by cytochrome C oxidase) [135] (Figure 3).

This method would modulate the activity of the malfunctioning neuroanatomical region of the CNS, in the case of GAD–DLPFC. The action itself would be associated with the stimulation of complex IV of the mitochondrial respiratory chain and increasing ATP levels, as well as acting through the release of calcium and activation of gene and transcription factor expression. Schiffer et al., on a small group of patients with depression, who were also assessed using the HAMA scale for anxiety levels, showed that 2 weeks after treatment, 6 out of 10 patients using tNIRS (4 sessions, including one to the left forehead, right forehead, and two placebos in the same places) experienced remission in the Hamilton rating scale for depression (HAM-D scale) (score below 10), and 7 of them in the HAMA scale [136]. In addition, increased blood flow associated with the procedure was observed in the frontal pole, measured using total hemoglobin (cHB) with NIR spectroscopy. This is consistent with assumptions about the impact of tNIRS on adenosine triphosphate (ATP) production. This flow may be indirect evidence of increased frontal cortex activity at the stimulation site. In Maiello et al.’s [137] study, the subjects were strictly individuals with GAD. Interestingly, not only did anxiety symptoms improve, but also sleep on the Pittsburgh Sleep Quality Index (PSQI) scale. However, this was a pilot study without a placebo. On the other hand, Wang et al. conducted a randomized study in which tNIRS 820 nm was used for 2 weeks, and EEG-TMS was performed before and after sessions to assess changes in conductivity. Stimulation was applied to the left DLPFC [138]. The researchers indicated improvement after using this method, which lasted for two months, measured by the HAMA scale, while EEG-TMS after the procedure showed improved DLPFC activity. Although evidence supporting tNIRS use is currently limited, it is difficult to draw definitive conclusions, mainly due to the small number of studies, participants, and the relatively short observation time of patients. It is possible that in the future, this method will have more applications and its effectiveness will be confirmed. At the moment, the relatively small number of studies limits the indication of which protocol would be best, as there are significant differences between the presented studies in terms of study duration, equipment, or duration of the entire therapy.

## 7. Electroconvulsive Therapy

Electroconvulsive therapy (ECT) is a relatively safe and one of the most effective methods of treatment in psychiatry. However, it seems that due to the necessary anesthesia for the procedure, the required anesthesiological assistance, and possible side effects, it is not the preferred method for treating anxiety disorders. Another important consideration in ECT is to assess the benefit-risk ratio, which, in the case of anxiety disorders, is rarely favorable: What is more, there is a lack of data that supports using ECT in anxiety disorders. In cases of anxiety disorders, the lack of evidence and possible risks associated with the procedure make it less favorable than other methods. In addition, Huang et al. indicated that the reduction of anxiety in the case of its co-occurrence with depression, is not as great as the improvement of depressive symptoms, moreover, it occurred later, which also suggests that the effect of ECT on both may be a separate pathway [139].

However, there is quite a bit of interest in this method for PTSD. Already in 2007, Watts indicated with his study, in which he treated depression with co-morbid PTSD, the effectiveness of ECT in reducing its symptoms as well [140]. Moreover, Araujo et al., [141] in their study, indicate that ECT can aid in the treatment of PTSD. They showed that PTSD patients who received traumatic memory before ECT treatments experienced a reduction in reactivity to that memory, as well as an anxiety reduction, after ECT. This is in line with the hypothesis quoted by Andrade et al. [142] who indicated that recalling traumatic events before treatment could lead to improvements in their perception and reactivity after treatment. In a randomized trial that treated PTSD co-existing with depression, there was no proven advantage of recalling traumatic memories over non-traumatic memories. The study found that ECT was effective in both groups, and the effects were observed three months after the study as well [143]. Other studies, conducted on a relatively small group of patients with PTSD, also provide support for the effectiveness of ECT: Youssef et al. study indicated that both low amplitude seizure therapy (LAP-ST) and standard right unilateral (RUL) ECT can be effective [144]. However, the number of available studies and clinical case reports is quite small. This limitation restricts our ability to draw definite conclusions [145]. The meta-analysis by Zhong et al. supports the view that studies to date are consistent in showing a potentially positive effect of ECT on PTSD [146], justifying further randomized trials.

## 8. Biofeedback

Biofeedback (BF) is a non-invasive treatment technique that combines physiological and psychological interventions to address psychiatric disorders. Essentially, individuals receive feedback signals (audio and/or visual displays) to enhance awareness of subconscious physiological activity and gain control over behaviors [147]. Various types of biofeedback exist, including electromyography biofeedback (EMGB), skin conductance biofeedback, and heart rate variability biofeedback (HRVB). However, neurofeedback (NF) is the most common, measuring brain waves to enhance neuromodulation and stabilization. In EEG-NF, data are recorded via electrodes placed on the scalp and displayed on a computer screen, enabling patients to visualize their mental state in wave frequencies [148,149] (Figure 4). 

Alpha (8–12 Hz) oscillations are well-established as indicators of sensory activity, exhibiting decreased levels in contexts of intrinsic sensory hyperactivity [150]. In essence, alpha waves are indicative of a state of relaxation, and their prevalence tends to rise as stress levels diminish [151]. 

Numerous studies have explored neuroanatomical abnormalities as markers in anxiety and fear-related disorders. Findings revealed decreased volumes in the left insula and lateral/medial prefrontal cortex and increased right putamen volume in GAD, while in fear-related anxiety disorders, less robust alterations were observed in the lingual gyrus with intact frontal integrity. These distinct neurobiological alterations may serve as therapeutic targets for disorder-specific neuromodulation interventions [152]. 

### 8.1. GAD

In a study by Hou et al., it was shown that neurofeedback, which was controlled by the projected recording of alpha waves, was an effective method for reducing anxiety, regardless of whether the EEG recorded signals from the right or left hemisphere (parietal lobe), with no significant differences between the groups. However, this study has notable limitations, such as the lack of a placebo group or the inclusion of patients who used other treatment methods, with all patients being women. Interestingly, the observed improvement measured by the state-trait anxiety inventory (STAI) scale was already evident at the first questionnaire completion (after five sessions) and even more pronounced at week 4. The relatively short observation period prevents any conclusions about whether NF maintains its therapeutic effect over a longer period [149].

Moreover, it appears that neurofeedback enhances stress control in stressful situations, such as musical performances [153], and improves mood and self-confidence in medical students [154]. However, these findings pertain to individuals without initial anxiety disorders, whose threat-related information processing is disrupted [155]. Research indicates that such individuals are more sensitive to threatening stimuli than to neutral ones, as seen in GAD [156]. Nonetheless, NF may support the functioning of such individuals by teaching them to reduce alpha waves. This aligns with the study by Dadashi et al., where NF was used to treat GAD in 14 patients, with another 14 receiving treatment after the study. The study demonstrated symptom reduction on the GAD-7 scale. However, this study also had significant limitations, such as comparing groups over time and lacking a placebo group. Additionally, the psychiatrist assessing the 28 patients post-intervention indicated that the study group did not meet the criteria for GAD diagnosis. Therefore, some caution is needed when interpreting these data. Diagnosing psychiatric conditions is a time-intensive process, and while the results are promising, we do not know if multiple diagnostic sessions would change the outcome. This is noteworthy because the researchers compared the effect before and after the NF treatment cycle, not its effect over time [157]. A potential solution to this issue might be the study by Rice et al., where NF aimed at increasing alpha waves significantly reduced autonomic arousal (measured by HR) even 6 weeks after an 8-session NF treatment. However, this was tested using psychophysiological tests in response to a stressor, which may not fully reflect daily functioning. Regardless, it is worth noting that this study also attempted placebo control, and other patients treated with methods such as EMG biofeedback or NF aimed at reducing alpha waves also benefited, though these benefits were not as long-lasting as those in the group targeted for increasing alpha waves [158]. Despite a recent meta-analysis not showing significant efficacy of biofeedback treatment compared to other methods focused on learning physiological parameter changes, and the specific effects of biofeedback being indistinguishable from non-specific treatment methods [159], it can still be a useful treatment form in selected cases. An example is the study by Schoneveld, where the MindLight game, using NF, taught children aged 7–12 to control their alpha wave levels. This was associated with navigating through a haunted house in the game, with the light based on alpha wave strength [160]. The main question was whether this method was more practical than CBT in reducing anxiety levels. The researchers suggested that a specific group of children who quickly become bored or discouraged might benefit more from this form of treatment, especially since MindLight and CBT were rated equally in terms of anxiety and difficulty.

Another form of biofeedback based on CNS changes is illustrated by the study by Scheinost et al., which used fMRI imaging of the orbitofrontal cortex to measure anxiety levels and used it in NF. The effects of learning to modulate this region were visible a week after NF, and the researchers do not rule out that this method may have permanently modified brain functionality. Using fMRI imaging seems to be an interesting topic as it allows learning to control a specific brain region in terms of its involvement in anxiety, for instance. This study showed that training-related fMRI changes correlated with a decrease in anxiety levels—in this case, specific anxiety—fear of contamination [161]. Additionally, Wang et al. also indicated that NF might be an effective method in combating the amplification of negative stimulation in individuals with high anxiety levels [162]. Therefore, NF could be a therapeutic component, particularly in individuals with high anxiety levels, for example, as support in personality disorder therapy.

There is also evidence of the effectiveness of other forms of biofeedback in GAD. Not only NF but also heart rate variability biofeedback may be useful in GAD. Park and Roth showed a clinical case of a patient who learned to control autonomic arousal measured by heart rate (HR) through diaphragmatic breathing control. This was an addition to the patient’s primary treatment. Interestingly, the patient maintained the improvement achieved through the training even 8 weeks after completing the 7-week protocol. These data show that this type of biofeedback might be a good idea to teach patients tension control and serve as a form of stress reduction [163]. On the other hand, Agnihotri in their study showed that EMG biofeedback, focused on assessing galvanic skin resistance, might even be more effective than EEG [164]. This suggests that various forms of biofeedback can be tailored to patient preferences.

Given the presented methods, it seems that NF aimed at increasing alpha waves has the most extensive research in GAD. However, this does not limit the use of biofeedback in other forms, such as EMG biofeedback or HR biofeedback. It should be emphasized that this method is unlikely to be a standalone therapy. In many studies, it is an additional tool, for instance, in CBT therapy [165]. It appears that the most beneficial use of such methods is in individuals with significantly elevated levels of autonomic nervous system tension, considering that its components are primarily what patients learn to regulate in the vast majority of biofeedback methods. However, there is a relatively small number of studies indicating how long the effects of these methods last. Another major problem in biofeedback research is the proper blinding of the control group, which is not always fully ethical and its implementation could lead to, for example, delaying treatment or worsening GAD symptoms in case of “false” EEG graphs and others.

### 8.2. SAD

Biofeedback has also been studied in cases of phobia, particularly social phobia. However, in the case of social phobia, biofeedback can sometimes lead to unfavorable situations, understood as fixation on a given parameter that biofeedback should teach to control. This is illustrated by the study by Gerlach et al., which showed that people with social phobia place more importance on visible anxiety symptoms than healthy individuals: using HR biofeedback, audible as sound, indeed reduced heart rate in a stressful environment between therapy sessions, but people with social phobia still exhibited high anxiety levels related to the fear of their tension/stress being visible [166]. Moreover, it seems that in the case of social phobia, it is important to prepare the patient for the process, such as presentations or speeches, as it allows correcting misinterpretations of information about being observed and improves performance under stress, as seen in the study by Nilsson et al.: individuals who received audio feedback (listening to recordings of their speeches) adapted better adapted and coped with the task when given cognitive preparation support. Logically, anxiety levels decreased in correlation with positive self-assessment of the patients’ speeches [167]. Additionally, Nilsson & Lundh pointed out in a subsequent study that the key component of cognitive preparation helping to reduce anxiety levels during speeches was reduced self-focus (RS) [168] and it should be considered in such learning forms. Thus, in the case of social phobia, it is most important to educate the patient and help them develop techniques to alleviate tension or self-focus, with selected biofeedback as a therapy element.

### 8.3. PTSD

In studies on PTSD patients using fMRI-supported NF, results showed that a shift in amygdala complex connectivity led to decreased arousal, increased resting alpha synchronization, and ultimately reduced PTSD symptoms [169]. Although changes appeared relatively quickly, temporal or momentary changes do not determine the duration of such procedures’ effects. Similarly, Polak et al. demonstrated the effectiveness of adding biofeedback in TF-CBT (trauma-focused CBT) in PTSD, showing that it helped achieve patient improvement faster, but the study was limited by a very small group of patients [170]. In another study, Lande et al. used HR variability (HRV) biofeedback as a complementary addition to PTSD treatment but did not show that biofeedback was more effective than traditional therapy [171].

In contrast, Zucker et al. who also used HRV, demonstrated improvement, although the duration of treatment in their case was longer by a week. However, in both studies, the patient group was rather small, and different software was used. Therefore, caution should be exercised when discussing the effectiveness of HRV biofeedback in this disorder. Reports indicate a relationship between HR variability and the severity of PTSD symptoms, but the evidence is quite limited, and significant methodological differences prevent us from drawing definitive conclusions [159]. 

### 8.4. Conclusions

Biofeedback appears to be an extremely interesting tool supporting the therapy of anxiety disorders. Due to the nature of social phobia and its symptoms, excessive focus on autonomic parameters may have the opposite effect. In this case, significant clinical experience is required, and current alternative treatment methods seem to be a better option. Unfortunately, the data from the literature carry a low level of evidence, as indicated by Tolin et al. [159]. This necessitates further research with better-designed control trials and continuous supervision of the procedure. Regarding this, it seems that the variety of biofeedback techniques allows for personalization and method selection according to patient preferences. Additionally, the possibility of exercising at home and the therapeutic process itself can bring tangible benefits to patients. Studies on biofeedback indicate that it may be most useful in GAD. However, it is important to note that studies do not indicate how long the effects of this method last. The fundamental question, therefore, remains whether biofeedback, apart from its high innovativeness, can actually compete with mindfulness, acute stress reduction methods, or CBT. It seems that its place in the treatment of anxiety disorders is mainly as a support to the main therapy.

## 9. Other Methods

Many other alternative methods deserve a separate thread to describe in detail, or the small amount of research on them makes it impossible to draw preliminary conclusions. Nevertheless, these methods may also prove to be alternatives to classical treatment in the future, so in this section, we have selected methods that seem particularly promising.

### 9.1. EMDR (Eye Movement Desensitization and Reprocessing) 

EMDR is a method with well-proven efficacy in the treatment of emotional trauma, which underscores its usefulness in PTSD [172]. Due to the amount of research and analysis, and EMDR’s well-established position, this method deserves a separate paper, so in our review, we decided to only briefly summarize recent reports by previous researchers. This method produces results in PTSD in the short term, as indicated by the Auren et al. study [173], and applied as one of the methods in intensive, short-term treatment of PTSD, the effects of treatment may persist in the long term. In the follow-up check of the Auren et al. study, the effect persisted even after 12 months [174]. However, it should be noted that EMDR was part of a composite therapy. However, according to the recommendations, EMDR is also very effective as a stand-alone treatment for PTSD, matching CBT [175], with the advantage of the therapeutic formula itself, which does not involve questioning the patient’s beliefs, detailed description of events, or prolonged exposure [176]. The Rasines-Laudes meta-analysis indicated that treatments that had more than 60 min per session, had more than 8 sessions, and had a professional therapist contributed to a significant reduction in symptoms [177]. However, both this analysis and earlier ones [178] point to a need for even better methodologically designed studies. A meta-analysis by Wright et al., however, indicated that the method was no more effective than other psychological methods. Interestingly, they showed that unemployed participants responded worse to therapy, and dropout was higher in men than in women [179]. Torres-Giménez et al. on the other hand, emphasize that this method can also be effective as an early intervention after a traumatic event [180].

### 9.2. VNS (Vagus Nerve Stimulation)

In clinical neurology and psychiatry, various nonpharmacological brain stimulation techniques have been used, for example, vagus nerve stimulation (VNS). The vagus nerve innervates the nucleus tractus solitarius on both sides, which is linked to brain regions responsible for regulating emotion, mood, and seizure activity [181]. Transcutaneous auricular vagus nerve stimulation (taVNS) is an effective noninvasive method for treating neuropsychiatric disorders. By targeting the auricular branch of the vagus nerve through electrical stimulation of the concha or the lower half of the back ear, taVNS can achieve modulatory effects comparable to those of invasive nerve stimulation (iVNS), which is associated with significant side effects and surgical morbidity [182].

Garcia et al. in a study on Twenty premenopausal women with recurrent MDD in an active episode found that exhalatory-gated RAVANS (novel respiratory-gated auricular vagal afferent nerve stimulation) was notably linked to heightened activation of the subgenual anterior cingulate, orbitofrontal, and ventromedial prefrontal cortices. It also showed increased connectivity between the hypothalamus and dorsolateral prefrontal cortex, as well as from the nucleus tractus solitarii to the locus coeruleus and ventromedial prefrontal cortex. These findings were associated with a positive impact on brain response to negative stressful stimuli, which leads to alleviating depressive and anxiety symptoms [183]. Moreover, in the treatment of panic disorder, additional vagal innervation seemed to ease panic attacks better than breathing exercises. For generalized anxiety, the vagal innervation method resulted in a reduction in psychic anxiety, with no significant changes in somatic anxiety symptoms [184]. 

A study by M Burger et al. aimed to determine whether transcutaneous stimulation of the vagus nerve (tVNS) could speed up extinction memory formation and retention in fear-conditioned humans. Results confirmed the promising role of this method, as tVNS accelerated explicit fear extinction learning. However, it did not enhance the retention of extinction memory after 24 h [185]. 

In contrast, in a study on people suffering from chronic worries, tVNS did not influence the IOR (inhibition of return), which is the phenomenon, where individuals tend to divert their attention away from previously attended locations to explore new ones, demonstrating an ‘inhibition of return’ to the initially attended area. Additionally, there was no link between resting levels of vagally mediated HRV and IOR [186]. However, according to one more study, it was demonstrated that short-term tVNS might alleviate spontaneously occurring negative thoughts in high worries, which offers intriguing insights into the potential effectiveness of tVNS as an intervention for worry-related psychological issues [187]. 

Vagus nerve stimulation is a method, which effectively modulates brain pathways associated with mood, anxiety, and stress responses. Transcutaneous auricular vagus nerve stimulation seemed to be an effective and safe option for, in particular, major depressive disorder. The advantages of the tVNS technique include its ability to significantly reduce acute depressive and anxiety symptoms and its non-invasiveness. Thus, it has the potential to improve the treatment of other neuropsychiatric disorders, yet further research is needed.

### 9.3. MDMA as a Therapeutic Adjunct to Psychotherapy

3,4-methylenedioxymethamphetamine (MDMA) is a psychoactive compound with structural similarities to classical stimulants and psychedelics, like amphetamine and hallucinogenic phenethylamines. The MDMA experience commonly involves increased arousal and perception, sociability, improved mood, and feelings of euphoria. Physiologically, MDMA acutely raises the heart rate, blood pressure, and body temperature, and generally disrupts homeostatic control, leading to insular disintegration, related to trait anxiety and acutely altered bodily sensation [188]. MDMA’s neurobiological effects are consistent with the clinical impressions about adding MDMA to psychotherapy. Several studies have indicated that MDMA-assisted psychotherapy is useful in overcoming PTSD and anxiety-related disorder [189]. 

In a study by Mithoefer et al., it was shown that MDMA significantly improved PTSD symptoms with no drug-related serious adverse events, adverse neurocognitive effects, or clinically significant blood pressure increases [190]. By combining data from six phase 2 trials, it was shown that active doses of MDMA with psychotherapy turned out to be effective and safe for both civilians and veterans/first responders with chronic PTSD who had not responded to pharmacotherapies or psychotherapy in previous trials. Moreover, results indicate a significant effect after two blinded active doses of MDMA for participants with highly refractory PTSD [190]. Another study was conducted to examine whether MDMA-assisted therapy is an optimal and responsible option for autistic adults with social anxiety, and the results were promising. MDMA seemed to rapidly improve social anxiety symptoms while being initially safe [191]. 

Several explanations exist for the effects of MDMA-assisted psychotherapy on anxiety and other symptoms. One potential mechanism for MDMA’s reduction of PTSD symptoms could involve its ability to decrease amygdala activity during exposure to negative stimuli while increasing frontal lobe activity, as traumas and emotions are addressed through similar neural mechanisms. Also, decreased right insular activity induced by MDMA may alleviate anxiety by reducing attention and concern about the bodily experience of anxiety. All in all, MDMA plays a promising role as an adjunct to psychotherapy for various anxiety-related conditions, especially PTSD. However, more studies with larger sample sizes and among more diverse populations need to be performed to fully understand the potential of this compound.

### 9.4. Cryotherapy

It is crucial to differentiate between two types of cryotherapy devices: whole-body cryotherapy (WBCT) and partial-body cryotherapy (PBCT). WBCT entails being inside a cryo-chamber where the air temperature drops between −50 °C to −150 °C. Conversely, PBCT is administered in cryo-saunas, resembling can-shaped barrel coolers, filled with air and liquid nitrogen mist, reaching temperatures around −190 °C. Notably, PBCT excludes the head from exposure, utilizes direct nitrogen vapor injection, and varies in size and portability compared to WBCT. Beyond their operational mechanisms, the distinction between WBCT and PBCT extends to the physiological response they elicit. PBCT subjects participants to a combination of cold and hypoxia stressors simultaneously, potentially activating unique cellular pathways compared to cold exposure alone [192]. 

WBCT offers various benefits including pain relief, muscle relaxation, enhanced muscle strength, and improved joint mobility. These effects persist for hours post-treatment, facilitating intensive rehabilitation. Additionally, WBCT triggers hormonal responses, elevating metabolism and plasma levels of adrenaline, noradrenaline, adrenocorticotropic hormone (ACTH), cortisone, pro-opiomelanocortin (POMC), β-endorphins, and male testosterone [193]. POMC is a precursor to critical biologically active compounds, which influence appetite regulation and sexual behavior. The role of α-MSH, derived from POMC, in appetite and sexual behavior regulation is particularly noteworthy. Considering the neurobiological theories of depression, which revolve around hypothalamic-pituitary-adrenal axis dysregulation, it is intriguing to note the involvement of brain opioid peptide systems in mood regulation, stress responses, and food intake control. Such a comprehensive systemic response triggered by cryotherapy suggests potential applications in treating mental disorders, such as anxiety disorders, underscoring its multifaceted therapeutic potential [194]. 

In a study conducted by Rymaszewska et al. in 2007 [195], similar to their 2008 [196] research, the effects of WBCT as an adjunctive treatment for depressive and anxiety disorders were investigated. The participants underwent 15 daily WBCT sessions, each lasting 120–180 s, with temperatures ranging from −110 °C to −160 °C. The study revealed significant improvements in psychopathological symptoms and life satisfaction among those receiving WBCT. On the anxiety scale (HARS), the experimental group showed more significant decreases than the control group. On the Hamilton Rating Scale for Depression (HDRS), the experimental group experienced a significant decrease.

Furthermore, on the life satisfaction scale (SSŻ), the experimental group had more significant increases. The results suggest that short-term exposure to extremely low temperatures via WBCT may be beneficial as an adjunctive therapy for affective and anxiety disorders, leading to significant reductions in psychopathological symptoms and improvements in life satisfaction. One significant limitation was the demographic composition of the sample, which consisted predominantly of female participants. This lack of gender diversity may limit the generalizability of the findings, as the effects of WBCT could differ between genders due to biological and psychological factors. Despite the small sample size and lack of randomized participant selection, these findings align with the 2008 study’s conclusions on the potential benefits of WBCT. Both studies indicate that WBCT might be an effective short-term treatment to enhance mood and well-being [195,196]. Szczepańska-Gieracha et al. studied the effects of WBCT on two groups: 34 individuals with spinal pain syndromes and 21 with peripheral joint disease. Each participant underwent 10 WBCT sessions at −100 °C, starting with shorter acclimatization sessions (1–2 min) followed by eight sessions of 3 min each. After treatment, participants reported significant improvements. WHOQOL-Bref (the World Health Organization Quality of Life: Brief Version) scores, measuring quality of life, showed a modest but positive increase. PGWBI (Psychological General Well-Being Index) scores, assessing well-being, indicated a more substantial improvement, reflecting better daily functioning. Interestingly, women showed a greater rate of improvement than men, and those with spinal pain syndromes improved more than those with peripheral joint disease. Participants with severe depressive symptoms at baseline saw the most significant gains, particularly in mood, overall well-being, and self-control [197]. 

#### 9.4.1. Temperature-Based Therapies in Anxiety-Provoking Medical Situations

Acupressure, a therapeutic technique rooted in traditional Chinese medicine, involves applying gentle pressure to specific points along the body’s meridians. Its fundamental aim is to regulate the blood flow and vital energy, known as qi. Through this practice, physiological coordination and balance are believed to be enhanced [198]. Research indicates that acupressure can have significant effects on neurotransmitter levels and reduce the levels of adrenocorticotropic hormones and 5-hydroxytryptamine (serotonin) [199]. By modulating these biochemical pathways, acupressure demonstrates promise in alleviating anxiety and promoting relaxation. Mirzaee et al. investigated the efficacy of acupressure with and without ice on the Hugo point (LI 4) during labor in women. The Hugo point, located between the first and second metacarpal bones, is associated with the large intestine energy channel. Participants had a mean age of 25.7 years, gestational age over 37 weeks, and approximately half had education above diploma level. The study utilized the visual analog scale for pain (VAS) to assess labor pain and the state-trait anxiety inventory (STAI) to evaluate anxiety levels. Their findings revealed that acupressure on the LI-4 point effectively reduces labor pain, regardless of whether ice is used, as measured by VAS. Both methods were more effective than the control group, with acupressure without ice showing greater efficacy than with ice. However, neither technique significantly impacted anxiety levels, according to STAI. Despite the clinically significant reduction in labor pain, there was no significant effect on anxiety [200]. 

In the realm of pediatric healthcare, vaccination often inflicts considerable pain on young patients, potentially fostering apprehension towards medical procedures and subsequent reluctance to seek medical care. One notable intervention explored in this context is the Buzzy device, which incorporates cold and vibration therapy. During the study conducted by Redfern et al., the Buzzy device was applied directly over the injection site for 30 s before being slightly repositioned and maintained throughout the needle stick procedure. The Wong–Baker Faces Pain Scale, and its modified version for assessing anxiety levels were used in the study. The study’s findings revealed a notable discrepancy in child-reported pain levels between the group that received the Buzzy intervention and the control group, with the former reporting significantly lower pain scores. Interestingly, while the Buzzy device effectively alleviated pain during vaccinations, it did not yield a significant impact on children’s pre-procedural anxiety levels or their expectations regarding pain intensity. Notably, younger children reported higher pain scores, underscoring age as the primary factor influencing pain perception, surpassing the impact of the Buzzy device [201]. In a study conducted by Aktaş et al., strategies to mitigate pain and anxiety associated with the removal of chest tubes (CTR) were examined. Patients often report inadequate pain management during this procedure, indicating a need for effective interventions to alleviate discomfort. Various interventions were evaluated, including cold therapy, music therapy, and lidocaine spray. Notably, patients undergoing CTR experienced comparable pain levels immediately after and 20 min post-removal across control and intervention groups, indicating that these interventions did not significantly affect pain perception, as assessed using the 10-cm vertical Visual Analog Scale (VAS). Concerning anxiety levels, the cold therapy group demonstrated statistically lower anxiety scores compared to the control, music therapy, and lidocaine spray groups 20 min after CTR. This suggests that cold therapy effectively reduced post-procedural anxiety, as measured by the state-trait anxiety inventory. However, the absence of a placebo group and consideration for individual patient factors limit the study’s findings [202]. 

Venipuncture, a routine medical procedure for diagnosis and treatment, often induces discomfort in patients due to pain and anxiety. A study by Hur et al. [203] explored various interventions to mitigate these issues. The study evaluated subjective pain, anxiety levels, and physiological responses like peripheral oxygen saturation and pulse rate across different intervention groups. Researchers administered three interventions: heat therapy, cold therapy, and thermal grill illusion (TGI) therapy, each for 10 s. Heat therapy involved 40–45 °C temperatures, while cold therapy utilized temperatures ranging from 0–10 °C. TGI therapy, on the other hand, involves the simultaneous application of both heat (40–45 °C) and cold (0–10 °C) for 10 s. 

Interestingly, while there were no significant differences in reported pain and anxiety levels among the groups, patient satisfaction varied notably. The group receiving cold therapy reported the highest satisfaction, suggesting a potential preference for this intervention method. However, the study had certain limitations. It focused on healthy adults with recent venipuncture experience, potentially limiting the generalizability of the results to a broader population. Additionally, the temperature settings of the thermoelectric element band were not individualized, indicating a need for personalized approaches in future research. In summary, while the study did not find significant differences in pain and anxiety reduction across the intervention groups, it highlighted the potential of cold therapy to enhance patient satisfaction. 

#### 9.4.2. Conclusions

The use of cryotherapy and temperature-related therapies appears promising in reducing substance use disorders, increasing life satisfaction, and improving mood and relaxation, both in psychiatric disorders and somatic situations/disorders such as childbirth or spinal cord injury. The authors emphasize the usefulness of integrating these therapies as adjuncts to standard treatments, whether pharmacological or psychotherapeutic. However, it is noteworthy that most studies share similar limitations, such as small control groups, lack of randomization or placebo groups, and no assessment of long-term effects. Additionally, poorly differentiated study groups make it difficult to generalize the findings to broader populations, highlighting the need for further research. As research progresses, whole-body cryotherapy (WBCT) may emerge as a valuable tool in managing mood and anxiety disorders, offering hope for more effective and holistic treatment approaches.

### 9.5. Hyperbaric Therapy

Clinical hyperbaric oxygen therapy involves immersing a patient’s entire body in a pressurized environment and administering 100% oxygen for a specified diagnosis and treatment period [204]. Typically, this therapy increases pressure to 2.0–2.5 atmospheres (ATA) and lasts 90–120 min per session, with the number of sessions depending on the specific condition being treated [205]. Mechanistically, preclinical studies suggest that hyperbaric oxygen therapy may mitigate oxidative stress, inflammation, and neural apoptosis [206].

In a study conducted by Harch et al., male military, either on active duty or recently retired, with an average of 2.8 years post-blast-induced traumatic brain injury (TBI), underwent 40 sessions of hyperbaric oxygen therapy (HBOT) at 1.5 ATA. HBOT is a medical treatment that utilizes oxygen at pressures higher than atmospheric levels to address various pathophysiological conditions. The treatment regimen involved sessions lasting 60 min, twice daily 5 days per week, for 29 days. These participants exhibited high scores on measures of post-concussion symptoms and PTSD. The study showed significant improvements in symptoms, cognitive function, and brain blood flow. Several outcome measures demonstrated significant improvements following HBOT, including neurological exam results, full-scale IQ, memory, attention, executive function, post-concussion symptoms, PTSD symptoms, depression, anxiety, quality of life, and self-reported level of functioning. However, the study had limitations, including the lack of confirmation of post-injury brain MRI results in some subjects, unblinded investigators, lack of a control group, and potential investigator bias and placebo effects. Despite these limitations, the authors argue that these factors will unlikely fully explain the observed improvements. In summary, the study suggests that 40 sessions of HBOT at 1.5 ATA over 1 month were safe and led to significant improvements in symptoms, cognitive function, and quality of life in military personnel with chronic blast-induced post-concussion syndrome and PTSD, with many of these improvements maintained at a 6-month follow-up [207]. 

The study conducted by Feng et al. aimed to assess the efficacy of hyperbaric oxygen therapy (HBOT) and psychotherapy in addressing depression, anxiety, nerve function, and daily activities among patients with incomplete spinal cord injury (ISCI). Three groups were established: the HBOT therapy group, the psychotherapy group, and the control group. Patients in the HBOT group underwent daily HBOT therapy sessions six times a week for eight weeks. Each session lasted 110 min, with chamber pressure reaching 0.2 MPa (2.0 ATA) for 30 min twice, separated by a 10-min interval. The psychotherapy group received supportive and cognitive behavioral therapy from psychiatrists and psychologists once daily, six days a week, for the same duration. The control group received medications to enhance circulation and facilitate nerve cell repair, in addition to routine SCI rehabilitation therapy once daily, six days a week, for eight weeks. The rehabilitation regimen included respiratory function training, bladder training, standing, joint motion, muscle, posture conversion, sit-up, and sitting balance training. Results from the study revealed that both HBOT therapy and psychotherapy significantly alleviated depression compared to the control group, as assessed by the Hamilton Depression Rating Scale (HAMD). However, there was no significant difference in depression improvement between the two interventions. HBOT therapy also led to a reduction in anxiety, as measured by the Hamilton Anxiety Rating Scale (HAMA), whereas psychotherapy did not demonstrate a significant impact on anxiety levels. Furthermore, HBOT therapy improved nerve function, as evaluated by the American Spinal Injury Association (ASIA) score, and activities of daily living, as assessed by the functional independence measure (FIM) score, compared to both psychotherapy and the control group [208].

#### Conclusions

Hyperbaric oxygen therapy (HBOT) has shown efficacy in reducing anxiety and improving nerve function and activities of daily living among patients with incomplete spinal cord injury (ISCI). Furthermore, it has led to significant improvements in symptoms, cognitive function, and quality of life in military personnel with blast-induced traumatic brain injury (TBI) and PTSD. Despite limitations, such as small sample sizes and potential biases, the findings suggest promising outcomes for HBOT therapy in addressing psychological and neurological symptoms in specific patient populations. However, further research involving larger and more diverse samples would be essential to validate these results and broaden their applicability to diverse patient populations.

## 10. Herbal Treatment 

### 10.1. Lavender Oil

Silexan, an essential oil for oral use derived from Lavandula angustifolia flowers, has demonstrated anxiolytic effects in patients with subthreshold anxiety disorders and GAD [209]. Woelk et al. investigated the efficacy of silexan compared to lorazepam. They found very similar results in both groups, as measured by two HAM-A sub-scores: “somatic anxiety” (HAM-A sub-score I) and “psychic anxiety” (HAM-A sub-score II). It suggests that Silexan may be effective in treating both the somatic manifestations and psychic anxiety associated with GAD [210]. In 2014, Kasper et al. found that Silexan (at doses of 160 mg/day or 80 mg/day for 10 weeks) was more effective in improving the mental condition and health-related quality of life in patients with GAD compared to placebo, and was at least as efficacious as paroxetine [211]. In 2017, researchers investigated the effects of different dosages of Silexan on GAD patients. The study results indicated that administering a dosage of 160 mg/day of Silexan led to improvements in both psychic and somatic symptoms of GAD.

On the other hand, reducing the dosage to 80 mg/day resulted in less pronounced beneficial effects, indicating that this dosage may represent the lower end of the therapeutic range for Silexan in treating GAD [212]. Furthermore, withdrawal symptoms and dependency associated with silexan use have not been observed, indicating that silexan can be safely discontinued without tapering even after long-term treatment [213]. While emerging evidence suggests that Silexan exhibits a robust and clinically significant anxiolytic effect, further research is needed to explore its broader potential [214].

### 10.2. Cannabidiol

Cannabidiol (CBD), a primary non-psychomimetic compound derived from the Cannabis sativa plant, has been hypothesized to be effective in treating SAD and GAD. Functional neuroimaging studies have indicated that CBD (at a dosage of 400 mg) reduces subjective anxiety levels. It also decreases ECD uptake in regions such as the left parahippocampal gyrus, hippocampus, and inferior temporal gyrus, while increasing ECD uptake in the right posterior cingulate gyrus. These changes contribute to the reduction of anxiety [215]. In a simulated public speaking test (SPST) involving 12 social anxiety patients who received CBD pretreatment (600 mg), researchers observed a significant reduction in anxiety, cognitive impairment, and discomfort during their speech performance. 

Additionally, there was a decrease in anticipatory speech alertness. These findings confirm the anxiolytic effects of CBD, which, in a single dose, can alleviate the fear of public speaking—the cardinal manifestation of SAD [216]. However, the results were not promising in a placebo-controlled test evaluating the effects of CBD (150, 300, or 600 mg) among college students. A single dose of CBD did not reduce test anxiety (TA) but instead induced self-reported test anxiety [217]. A systematic review of numerous studies demonstrates that CBD does not induce significant increases in positive, negative, general, and total symptoms, unlike tetrahydrocannabinol (THC). There were also conflicting results regarding CBD’s ability to moderate the induction of psychiatric symptoms by THC, as hypothesized [218].

### 10.3. Kava Kava

Kava (Piper methysticum) is a South Pacific plant-based medicine known for its potential use as an anxiolytic due to its psychoactive constituents, the kavalactones. These compounds act on various neurochemical pathways, including reducing the neuronal reuptake of noradrenaline and dopamine, blocking voltage-gated sodium ion channels, and decreasing excitatory neurotransmitter release through calcium ion channel blockade. Particularly noteworthy is its effect on gamma-aminobutyric acid (GABA) response [219]. A study aimed to examine GABA concentration in the dorsal anterior cingulate cortex (dACC) in patients with GAD to confirm the efficacy of Kava as an anxiety treatment. Using proton magnetic resonance spectroscopy, researchers found that Kava modulated GABA levels in the brain region, with its concentration appearing to correlate with anxiety symptoms in GAD [220]. Another study compared the effectiveness of Kava-Kava (120 mg/day) with Opipramol and Buspirone in GAD. The results indicated that Kava-Kava was as effective and well-tolerated as the other medications [221]. In contrast, Sarris et al. found that Kava was ineffective in short-term treatment for diagnosed generalized anxiety but did not deny its efficacy in managing non-clinical situational anxiety and stress [222].

### 10.4. Valeriana

Valerian extract (Valeriana officinalis) is commonly used in some countries due to its hypnotic-sedative effect. Valepotriates, the main active components of valerian extract, are believed to have an anxiolytic impact on the psychic symptoms of GAD. However, due to limitations such as a small sample size, researchers require more supportive research [223]. Muller et al. reported that adding valerian extract to St. John’s wort (an antidepressant) can provide additional benefits among patients suffering from depressive and anxiety disorders. The combination therapy showed improvements, especially in anxiety symptoms such as tension/dysphoria, anxiety/nervousness, and sleep disorders [224].

Furthermore, a study on volunteers experiencing psychological stress showed that administering valerian root extract (300 mg/day) can increase alpha coherence in the frontal region of the brain. This increase is positively associated with neurophysiological integration, cognitive flexibility, and information processing. Although these changes confirm that valerian extract eases anxiety, researchers need to conduct more studies due to the negative psychological scale findings [225]. Another study demonstrated that a single oral dose of valerian extract, but at a higher dose of 900 mg/day, affects motor cortex excitability with a significant reduction in ICF (intracortical facilitation), which returns to baseline 6 h after administering valerian extract. This effect on ICF might be explained by allosterically modulating GABA receptors with valerian extract, leading to anxiolytic-like activity [226].

### 10.5. Chamomile

Chamomile (*Matricaria chamomilla* L.) is among the ancient herbs used for medicinal purposes due to its anxiolytic and hypnotic effects. Some studies have revealed improvements in sleep quality and relief of GAD symptoms after 2 and 4 weeks of chamomile consumption [227]. In long-term treatment for GAD, it has been established that chamomile is safe and reduces moderate-to-severe GAD symptoms. However, a lower risk of relapse has not been demonstrated [228]. According to a study by Keefe et al., patients with GAD who received chamomile experienced increased morning salivary cortisol levels, which are associated with a steeper diurnal cortisol slope after awakening. It is correlated with a significant reduction in anxiety symptoms [229]. However, another study concluded that not only does chamomile itself have a therapeutic effect, but high expectancy for positive outcomes can also play a role. Results showed significant improvements in anxiety symptomatology among participants with higher expectancy and more side effects when they expected them. Thus, treatment expectancies might play a crucial role in the efficacy of active medication, which applies to most herbal drugs [230].

### 10.6. Chamomile

The increasing volume of international studies, reviews, and meta-analyses suggests a significant interest in preparations containing, especially lavender oil, but also cannabidiol, Kava Kava, valeriana, chamomile, and even more. Although the described herbs were generally safe and notably reduced moderate-to-severe GAD symptoms, there are a few limitations we should take into account. The limited sample size and preliminary/exploratory nature of most of the studies contributed to the non-significant primary outcome findings.

Overall, while these natural remedies show promise, further studies, involving larger patient cohorts, higher fixed dosages, prolonged durations of drug administration, diverse methods of patient recruitment, and more specific evaluations of psychic symptoms, are needed to fully understand the potential and optimal usage of herbs.

## 11. Conclusions

Of the presented methods, MBI and EMDR seem to provide alternative treatments for anxiety disorders. However, these and other methods can also be adjuncts to classical treatments, with a particular focus on physical activity or the use of VR technology to support CBT. In the case of GAD, MBSR may initially be more effective in reducing symptoms than CBT, so it is worth combining these methods. It should be noted that for other methods and VR, most studies do not report clear results indicating additional benefits of their use.

Regarding physical activity or other methods presented here, it should be emphasized that they are not causal treatments. They do not achieve the effectiveness of CBT, but studies clearly show that they are useful adjuvant therapies. Physical activity reduces overall anxiety levels, seizure frequency, and intensity. Physical activity may allow the patient to feel increased control over their life. However, it is important to note that according to this theory, both high- and low-intensity exercises would be equally beneficial, which is not the case. Research indicates that physical activity is particularly effective when using resistance and high-intensity training for about 30 min, with repetition every 2–3 days. Studies do not conclusively establish how long improvement persists after exercise because it is used as an adjunct to other primary treatments. The literature presented above supports the idea that just a few training sessions allow improvements in the patient’s condition to be noticed, but the improvements do not seem to be induced and are maintained spontaneously without further repetitive exercises. In the case of VR technology, on the other hand, trials have been most promising for social anxiety during CBT. By playing out different scenarios, this method allows different exposures to be practiced in a comfortable environment for the patient. Another example of the potential use of VR is exposure therapy in the treatment of PTSD, but the current research is conducted on a small group of patients. Whether VR technology will be permanently integrated into psychotherapy practice in the future remains an open question. The main task of future research will be to unequivocally determine whether its use increases the overall effectiveness of CBT therapy. This is essential, as this method is associated with side effects, and without certain efficacy, it is not beneficial to use it. In the case of TMS, and BF, there are too little data from the literature to determine their real therapeutic effect. Other mentioned methods such as cryotherapy or MDMA-assisted psychotherapy, should also be used, but further research is required.

Herbal preparations remain a separate issue—the increasing use of herbs in Western countries means that when prescribing conventional drug treatment, it is necessary to establish possible interactions and clarify information about them from the patient. Regardless, herbal products can provide relief to a large group of patients, but their dosage is best determined on an individual basis due to, for example, changes in liver metabolism or the fact of complex herbal preparations. While herbs are generally safe and noticeably alleviate moderate-to-severe GAD symptoms, it is important to remember the significant limitations in their research. A large proportion of the studies do not take a placebo into account and are conducted on a small group of subjects, making it necessary to maintain subdued enthusiasm when talking about the effectiveness of such preparations. 

It is crucial to note that the conclusions drawn from the studies discussed primarily involve patients in stable conditions. Therefore, these findings should not be generalized to patients with severe conditions. Using these methods with proper clinical oversight is advised, as their use without appropriate pharmacological care could worsen the patient’s condition or prolong therapy. Further research is essential to establish the efficacy and safety of these approaches and to determine their role in enhancing overall treatment outcomes for anxiety disorders. Figure 5 shows an analysis of the benefits and risks associated with alternative methods, and after that, we present the key findings of our work.

Alternative therapeutic approaches are not only more cost-effective but also less prone to causing side effects, rendering them appealing treatment options.Alternative methods can have potential as adjuncts to traditional therapies, especially physical exercise and VR technologyMBI and EMDR seem to have the most significant therapeutic potential, providing standalone treatments for anxiety disorders. Although herbal products are generally safe and beneficial in reducing GAD symptoms, it is important to establish possible interactions with conventional medicationsFurther research is crucial to confirm the effectiveness and safety of alternative methods and to define their roles in improving overall treatment outcomes for anxiety disorders.

### Methods Qualified to Assist Classical Treatment

Alternative methods, though not without reason, can have the potential as adjuncts to traditional therapies. Of all the methods described, physical exercise seems to possess the most adjunctive qualities (Figure 6).

Resistance exercise training showed promising antidepressant effects, indicating its potential as a primary or supplementary treatment for moderate depressive symptoms. Combining yoga with CBT sped up the reduction of anxiety and depressive symptoms more effectively than CBT alone, with effects lasting up to 3 months. For those with a poor response to medication, Sudarshan Kriya Yoga is considered a dependable supplementary or alternative treatment. Moreover, yoga served as a helpful addition to cancer treatment, in decreasing anxiety, depression, and fatigue symptoms as well. 

The use of virtual reality (VR) for exposure therapy appears to be a promising option for patients dealing with anxiety disorders related to specific situations. Moreover, in agoraphobia, VR exposure therapy has the potential to be used as an alternative to CBT instead of just a supplementary treatment. Furthermore, researchers found that VRelax quickly reduced anxiety and depression, with stronger effects in patients receiving outpatient psychiatric treatment, suggesting VR’s capability as a supplement to medication.

When it comes to MBIs, in patients with anxiety disorders who previously underwent CBT, MBCT resulted in a greater reduction in subjective anxiety compared to those receiving CBT for relapse prevention (CBT-RP), but it was not sustained in the long term. 

TMS also seemed to be a good adjunctive method in the treatment of GAD. Studies indicate that TMS is frequently utilized when pharmacological treatment alone is insufficient and additional therapy is required.

Biofeedback seems to be a highly intriguing tool for supporting the therapy of anxiety disorders, primarily serving as an adjunct to the main treatment. For example, breathing biofeedback, when used alongside exposure in cognitive behavioral therapy, shows promise in accelerating the reduction of PTSD symptoms.

MDMA appears to be a promising addition to psychotherapy as well, for various anxiety-related conditions, particularly PTSD. However, further studies with larger sample sizes and more diverse populations are needed to fully understand its potential. Similar limitations occur in the case of WBCT research. Future research may benefit from closer examination of WBCT as an adjunctive treatment for depressive and anxiety disorders. Currently, findings indicate that short-term exposure to extremely low temperatures through WBCT may be beneficial as an adjunctive therapy for affective and anxiety disorders, significantly reducing psychopathological symptoms and enhancing life satisfaction. 

## Figures and Tables

**Figure 1 diseases-12-00216-f001:**
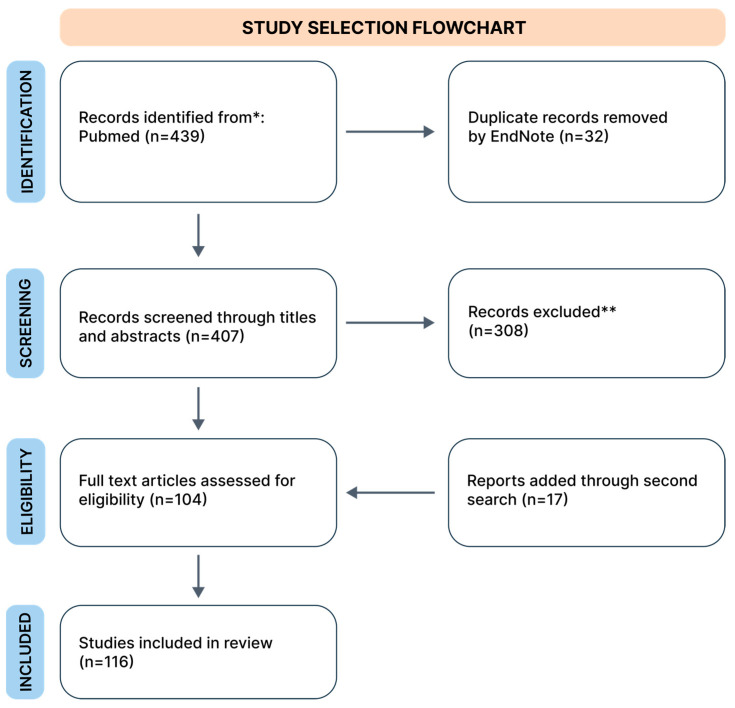
Flowchart of search strategy. * Due to frequent gaps in the PubMed database regarding selected forms of treatment, we included older studies if, after consultation between two authors, the study was deemed crucial (both authors indicated “yes” during the inclusion voting (“yes” or “no”). ** In this step, despite excluding studies that did not meet our inclusion criteria, we removed duplicate studies missed by the automated tool.

**Figure 2 diseases-12-00216-f002:**
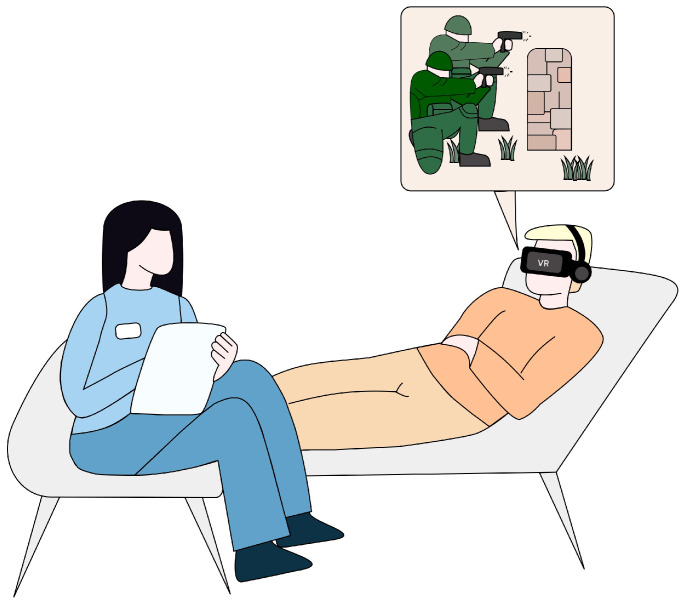
A visual example of the use of VR-supported exposure therapy for PTSD.

**Figure 3 diseases-12-00216-f003:**
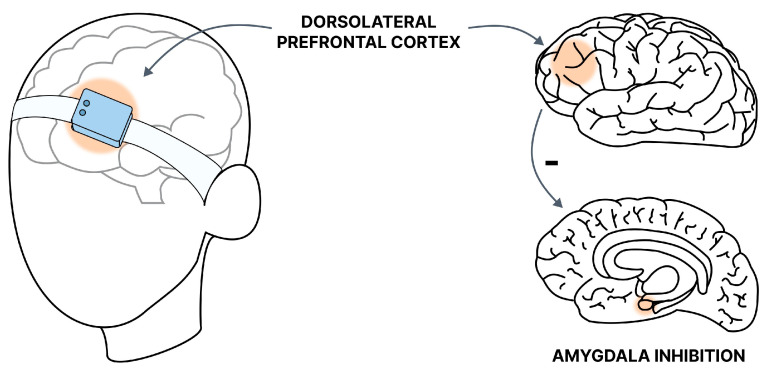
Graphic presentation of the tNIRS method. The target site of near-infrared stimulation in GAD is the DLPFC. The stimulation allows for increased inhibition of the amygdala and reduction of symptoms.

**Figure 4 diseases-12-00216-f004:**
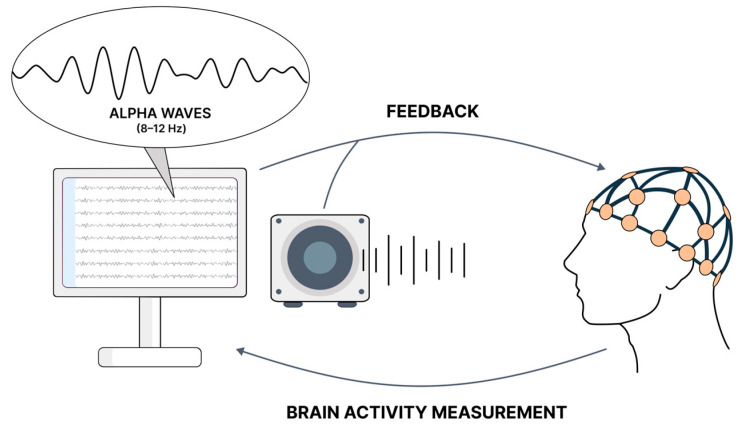
Graphic illustration of neurofeedback. The patient learns to control alpha waves in a range suitable for deep relaxation. When reaching or going beyond a certain measured parameter, the patient is informed by an audible signal or a visible change in the color of the alpha waves on the EEG monitor.

**Figure 5 diseases-12-00216-f005:**
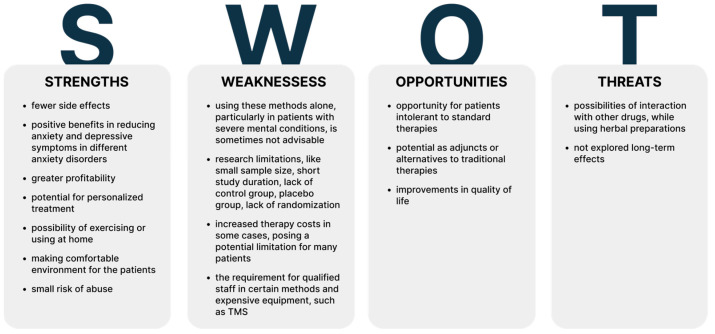
SWOT Analysis.

**Figure 6 diseases-12-00216-f006:**
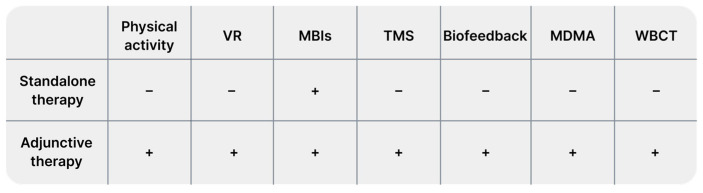
The role of selected alternative methods in the treatment of anxiety disorders.

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
