# Peer review of "Beyond Pharmacology: A Narrative Review of Alternative Therapies for Anxiety Disorders"

_diseases, 2024, doi:10.3390/diseases12090216_

Round 1

Reviewer 1 Report

Comments and Suggestions for Authors

This review summarizes evidence concerning the use of alternative treatment strategies in anxiety disorders. These conditions, despite being highly prevalent, often receive suboptimal treatment and impact heavily quality of life of affected people. Despite this topic require attention and the paper could thus be of great interest to a broad audience, there are major points that should be addressed before considering this review for publication.

-In the Introduction, Authors mention vulnerable categories of patients, in particular pregnant women. I would suggest to expand considering further populations, e.g., could alternative treatment strategies be useful for treating anxiety in adolescents or elderly people?

-    - This is a narrative review and Authors clearly state this in the Methods section. Anyway, I would suggest to explain the inclusion criteria for better clarifying how studies were selected;

-  - When presenting results from the included research, I would suggest to include the rationale for the effect of specific interventions, e.g., yoga, so that their effect on anxiety symptoms could be clearer to the reader - in this present version of the review, mechanisms of action are included only for some interventions;

-  -  When presenting results on physical exercise, consider elaborating on the limitations of the studies, particularly the demographic characteristics of the sample populations. For example, discuss how the predominantly White and well-educated sample might impact the generalizability of the findings and suggest areas for further research;

-      - Please, be clearer when introducing “control groups” in the study description, possibly mentioning their composition and which medications they assumed;

-          - In sections where long-term effects were not sustained (e.g., yoga interventions), provide potential reasons for this and suggest strategies for improving long-term outcomes, such as booster sessions or integrating social components;

-          Consider expanding on why mindfulness interventions might be particularly effective for generalized anxiety disorder, especially in contrast to other anxiety disorders;

-   -        I recommend providing a brief explanation of the differences between MBSR and MBCT when they are first introduced to highlight their unique approaches;

-         -  I recommend adding a brief explanation of the differences between TMS, rTMS, and tDCS early in the text to help differentiate these techniques;

-     -      It might be beneficial to include a brief explanation of why ECT is not the preferred method for treating anxiety disorders, beyond the need for anesthesia and potential side effects;

-  - I would suggest to simplify the discussion on herbal preparations, emphasizing the importance of individualized dosing and caution due to potential interactions with conventional medications;

-    -  I would recommend a more clearly differentiation between the fields where the various therapies discussed in the review could be useful, emphasizing the specific contexts in which each might be most beneficial;

-  -  I suggest strengthening the concluding remarks by summarizing the key findings more concisely and reiterating the importance of further research to validate the efficacy and safety of these alternative therapies, so that clinicians could benefit from a clear take-home message.

Some minor points are those concerning formal aspects as follows:

-          I would suggest using "substance use disorders”  instead of “ substance dependencies”

-          Please, ensure that all acronyms (e.g., GAD, CBT, PTSD, AGAD, SAD, PD, etc.) are spelled out and specified when introduced in the text for the first time;

-          Please, maintain consistency in terminology throughout the text. For instance, if it is use "resistance exercise training (RET)" initially, continue using "RET" instead of alternating with other terms like "resistance exercise”;

-      I would suggest to extensively review the manuscript for more accurate terminology in different parts of the text. Here are some examples: I would recommend using "prolonged" instead of "extended" when discussing the duration of medication use; it might be more accurate to replace "self-interruption" with "self-discontinuation" to describe the action of patients stopping their medication on their own; it could be helpful to rephrase "calming the HPA axis" as "regulating the HPA axis", etc.

Comments on the Quality of English Language

-      I would suggest to extensively review the manuscript for more accurate terminology in different parts of the text. Here are some examples: I would recommend using "prolonged" instead of "extended" when discussing the duration of medication use; it might be more accurate to replace "self-interruption" with "self-discontinuation" to describe the action of patients stopping their medication on their own; it could be helpful to rephrase "calming the HPA axis" as "regulating the HPA axis", etc.

Author Response

Dear Reviewer,

Thank you very much for your response and for your detailed comments, which will certainly improve the quality of our work. Please see the attachment. I will try to answer to your comments:

  •      I would suggest to extensively review the manuscript for more accurate terminology in different parts of the text. Here are some examples: I would recommend using "prolonged" instead of "extended" when discussing the duration of medication use; it might be more accurate to replace "self-interruption" with "self-discontinuation" to describe the action of patients stopping their medication on their own; it could be helpful to rephrase "calming the HPA axis" as "regulating the HPA axis", etc. --> We have reviewed the entire manuscript and changed all the terms we found to more relevant to the medical language. Thank You for this comment.
  • Some minor points are those concerning formal aspects as follows:

-          I would suggest using "substance use disorders”  instead of “ substance dependencies”

-          Please, ensure that all acronyms (e.g., GAD, CBT, PTSD, AGAD, SAD, PD, etc.) are spelled out and specified when introduced in the text for the first time;

-          Please, maintain consistency in terminology throughout the text. For instance, if it is use "resistance exercise training (RET)" initially, continue using "RET" instead of alternating with other terms like "resistance exercise”;

     I would suggest to extensively review the manuscript for more accurate terminology in different parts of the text. Here are some examples: I would recommend using "prolonged" instead of "extended" when discussing the duration of medication use; it might be more accurate to replace "self-interruption" with "self-discontinuation" to describe the action of patients stopping their medication on their own; it could be helpful to rephrase "calming the HPA axis" as "regulating the HPA axis", etc. --> We reviewed the entire manuscript again. We tried to find all the mistakes and corrected them. Thank you for examples, it makes our work easier.

  • -    -  I would recommend a more clearly differentiation between the fields where the various therapies discussed in the review could be useful, emphasizing the specific contexts in which each might be most beneficial;- 
    -  
    I suggest strengthening the concluding remarks by summarizing the key findings more concisely and reiterating the importance of further research to validate the efficacy and safety of these alternative therapies, so that clinicians could benefit from a clear take-home message. --> Of course, we improved the conclusions and added key points at the end of our work.
  •  I would suggest to simplify the discussion on herbal preparations, emphasizing the importance of individualized dosing and caution due to potential interactions with conventional medications --> Thank you for that information. We have simplified the summary on herbal treatment

  •   It might be beneficial to include a brief explanation of why ECT is not the preferred method for treating anxiety disorders, beyond the need for anesthesia and potential side effects; --> I toned down the chapter on ECT a lot, pointing out other aspects, and the risk-benefit ratio. This method is unique, but it is difficult to say whether many psychiatrists will like it applied on themselves.
  •      -  I recommend adding a brief explanation of the differences between TMS, rTMS, and tDCS early in the text to help differentiate these techniques; --> Thank you for letting us know about your experience with the chapter. I have corrected it and added definitions at the beginning of the chapter. I hope it is more readable now.
  •  I recommend providing a brief explanation of the differences between MBSR and MBCT when they are first introduced to highlight their unique approaches;  Consider expanding on why mindfulness interventions might be particularly effective for generalized anxiety disorder, especially in contrast to other anxiety disorders;--> We tried our best to cover this issue, thank you for this comment.
  •  Please, be clearer when introducing “control groups” in the study description, possibly mentioning their composition and which medications they assumed; --> We tried to describe the control groups to make it easier for readers to draw conclusions about the quality of the studies.
  • When presenting results on physical exercise, consider elaborating on the limitations of the studies, particularly the demographic characteristics of the sample populations. For example, discuss how the predominantly White and well-educated sample might impact the generalizability of the findings and suggest areas for further research; --> We added a discussion of the limitations of the study, focusing on the demographic characteristics of the study population. Thank you for comment.
  • When presenting results from the included research, I would suggest to include the rationale for the effect of specific interventions, e.g., yoga, so that their effect on anxiety symptoms could be clearer to the reader - in this present version of the review, mechanisms of action are included only for some interventions; --> We went through the entire manuscript and added the mechanism of action where we thought it was pertinent. In the case of “other methods,” due to the small amount of evidence, we decided not to add them. We hope it will be enough to cover this comment. 
  • This is a narrative review and Authors clearly state this in the Methods section. Anyway, I would suggest to explain the inclusion criteria for better clarifying how studies were selected; --> I tried my best to describe our search protocol. However, after trying to access all the data information, it seems that I am not able to describe it more precisely. My apologise for that.
  • In the Introduction, Authors mention vulnerable categories of patients, in particular pregnant women. I would suggest to expand considering further populations, e.g., could alternative treatment strategies be useful for treating anxiety in adolescents or elderly people? --> We have added a paragraph and addressed this issue. Thank you for your comment.

We would like to sincerely thank you for your time and comments, which allow us to objectively look at the main problems and weaknesses of our work. We believe that thanks to the response to your comments, our work has raised its level.

Kind regards,

Stefan Modzelewski

Reviewer 2 Report

Comments and Suggestions for Authors

This is a clinically useful paper.

It would be better if the following were revised.

1. In addition to the treatment methods described, it would be better if cognitive therapy, behavioral therapy, supportive psychotherapy, and insight-oriented psychotherapy were also covered.

2. 2. It would be better if it were mentioned that posttraumatic stress disorder is classified separately in the DSM-5 classification and that social anxiety disorder does not classify subtypes.

Author Response

Dear Reviewer,

Thank you for your time and precise comments. We have discussed them all and revised our manuscript. Please see the attachment.

2. 2. It would be better if it were mentioned that posttraumatic stress disorder is classified separately in the DSM-5 classification and that social anxiety disorder does not classify subtypes. --> Of course, we added information about classification of PTSD in DSM-V, and mentioned SAD in methodology section. Thank you for that comment.

1. 1 1. In addition to the treatment methods described, it would be better if cognitive therapy, behavioral therapy, supportive psychotherapy, and insight-oriented psychotherapy were also covered. --> Thank you for your valuable suggestion regarding the inclusion of individual psychotherapeutic methods. In response to the suggestion, we did a literature revision, to find some information about it. However, we found that there were insufficient materials, and we weren't able to make clear conclusions. Taking that into consideration, we had a discussion and decided to mention their possible integration within the broader methods more thoroughly analyzed in the text. This decision was made to maintain the structure, which primarily compares the adherence of well-studied CBT therapy to other methods where applicable. In the manuscript, we have outlined the need for further exploration of the topic.

Best regards and thank you for your valuable time,

Stefan Modzelewski on behalf of coauthors

p.s. I update the corrected version of manuscript, in the first answer i made a mistake and upload older version of our work. My apologise.

Reviewer 3 Report

Comments and Suggestions for Authors

Beyond pharmacology : narrative review of alternative therapies for anxiety

Sincere, I consider the manuscript is an excellent work. The paper can be published.
However, should be checked some minor errors

1)    In page 3, line 118

by calming the HPA axis
As a suggestion, please write the acronym of HPA

2)    In page 4, line 162

designed according to WHO and ACSM guidelines, significantly improved
Please write the acronym of ACSM

3)    In page 7, line 293

As a suggestion, in sub-titles write the complete word of each acronym
3.3. PTSD

4)    In page 12, line 546

As a suggestion, in sub-titles write the complete word of each acronym
 4.3 GAD

5)    In page 15, line 682

As a suggestion, in sub-titles write the complete word of each acronym
 5.3 GAD

6)    In page 15, line 725

As a suggestion, in sub-titles write the complete word of each acronym
 6. TMS

7)    In page 16, line 775
As a suggestion, in sub-titles write the complete word of each acronym
 6.2 GAD

8)    In page 18, line 860
As a suggestion, in sub-titles write the complete word of each acronym
 6.4. PTSD

9)    In page 21, line 998
As a suggestion, please write the complete word of acronym: TIAS scale

Author Response

Dear Reviewer,

I am grateful for such positive review. Me and my coauthors are pleased that you consider our work excellent. Thank you for your work and time.

We corrected all errors (1-9), which you find. Thank you for help in improving our paper. In case of 9) we found that acronim was written uncorrectly and we have changed it. Please see the attachment. All changes should be visible in added comments.

Best regards,

Stefan Modzelewski and coauthors

Round 2

Reviewer 1 Report

Comments and Suggestions for Authors

The manuscript appears substantially improved after Authors' revisions. I have no further comments.

Comments on the Quality of English Language

English language is significantly improved. Some typos can be checked during prooofreading.